



# Diffusion based modelling of combined temperature and moisture effects on carbon fluxes of mineral soils

Fernando E. Moyano[1]*, Nadezda Vasilyeva[2,3], Lorenzo Menichetti[4]

[1]University of Goettingen, Bioclimatology, Göttingen, 37077, Germany

[2]UPMC-CNRS-INRA-AgroParisTech, UMR 7618, Bioemco, Thiverval-Grignon, 78850, France

[3]V.V. Dokuchaev Soil Science Institute, Moscow, Russia

[4]Sveriges Lantbruksuniversitet (SLU), Ecology Department, Ulls Väg 16, 756 51 Uppsala, Sweden

10    *Correspondence to*: Fernando E. Moyano (fmoyano@uni-goettingen.de)





**Abstract.** While $CO_2$ production in soils strongly responds to changes in temperature and moisture, the magnitude of such responses at different time scales remains difficult to predict. In particular, little is known of the mechanisms leading to interactions in the effects of these drivers on soil $CO_2$ emissions, even though such observations are common. Here we compare a number of modelling approaches to test which underlying mechanisms best simulate the interactive responses measured in

soils incubated under combined levels of temperature and moisture. We found that two model components were critical for reproducing the observed interactive patterns: 1. Michaelis-Menten reaction kinetics, which strongly improved the model fit when applied to decomposition reactions, and 2. diffusion of dissolved C and enzymes. The latter replaces conventional empirical functions as a mechanism relating moisture content with C fluxes. Indeed, empirical functions failed to capture the main observed interactions. After model calibration we were able to explain 84 % of the variation in the data. Model

simulations resulted in a decoupling of decomposition and respiration C fluxes in the short and mid-term, with interaction effects and general sensitivities to temperature and moisture being more pronounced for respiration. Sensitivity to different model parameters was highest for those affecting diffusion limitations, followed by activation energies, the Michaelis-Menten constant, and carbon use efficiency. Model validation resulted in a high fit against independent data ($R^2 = 0.99$). The same underlying model parameters resulted here in different apparent temperature sensitivities compared to the calibration step,

demonstrating a strong effects of initial soil conditions. With these results we could demonstrate the importance of model structure and the central role of diffusion and reaction kinetics for simulating complex soil C dynamics related to temperature and moisture interactions. Future studies should further validate this mechanistic approach and extend its use to a larger range of soils.

## 1.        Introduction

Soils are a main component of the global carbon (C) cycle, storing ca. 2200 Pg of C in the top 100 cm alone according to recent estimates (Batjes, 2014). This soil C pool is dynamic, and often exists in a non-equilibrium state as the result of an imbalance between input and output C fluxes, in which case it will act either as a C sink or source over time. Changes in the speed at which soil organisms decompose soil organic matter (SOM) and mineralize soil organic carbon (SOC) into $CO_2$ are one way in which an imbalance can occur, producing a net sink or source of atmospheric $CO_2$.

It is well known that SOC mineralization and resulting $CO_2$ fluxes are highly sensitive to variations in soil temperature and moisture (Hamdi et al., 2013; Moyano et al., 2013). As a result, feedbacks effects, either positive or negative, are expected to occur from the interaction between climate change and global soil C stocks (Crowther et al., 2016; Davidson and Janssens, 2006; Kirschbaum, 2006). However, the direction and magnitude of such feedbacks at the global scale remain uncertain. Increased soil respiration with a resulting net loss of soil C, and thus a positive climate feedback, is expected with the warming

of permafrost soils and the drying of wetland soils. But there is still much uncertainty and a lack of consensus regarding the long term response to climate variability of soils that are non-saturated, non-frozen, and dominated by a mineral matrix (Crowther et al., 2016), i.e. soils found under most forests, grasslands and agricultural lands.



Future predictions of soil C dynamics require the use of mathematical models. Early soil C models, and most still in use, are based on first order decay of multiple C pools, with temperature and moisture having a general non-interactive effects on decay rates (Rodrigo et al., 1997). These models, when appropriately calibrated, do well at simulating soil respiration fluxes of soils under relatively stable conditions. They were often developed to approximate long term steady state conditions under specific land uses. They are also capable of fitting long term trends of soil C loss, such as data from long-term bare fallow where all litter input has stopped (Barré et al., 2010). However, they lack a theoretical basis justifying their basic assumptions of pool partitioning and decay mechanisms. They also generally need calibration for specific soil types or land cover types, and often fail to properly simulate observed short and mid-term variability in soil respiration.

Some of the most relevant observations these models have failed to reproduce include: changes (typically a dampening) of temperature sensitivities of decomposition over time (Hamdi et al., 2013), non-linear responses to soil moisture content (Borken and Matzner, 2009), and changes in decomposition rates in response to variations in concentrations of organic matter (Blagodatskaya and Kuzyakov, 2008). Such model shortcomings, which reflect missing or wrongly simulated processes, create a difficult to quantify uncertainty in global long term predictions of soil C and its feedback to climate change. It is therefore unclear if first order models can predict long term changes in C stocks under more dynamic (and therefore realistic) environmental conditions.

Second order models have a more realistic basic structure compared to conventional first order models, since they simulate organic matter decomposition as a reaction between two pools, one of these being the decomposer pool (a microbial pool or enzyme pool). This single but fundamental change in decomposition kinetics strongly affects predicted long term changes in soil C, largely as a result of the dynamics of the decomposer pool, which itself can respond to temperature in a number of ways (Wutzler et al., 2011). Second order models also lead to more complex dynamics of short to mid-term soil respiration, with apparent temperature sensitivities that vary over time, more in line with many observations.

The temporal variability in the response of decomposition to moisture is most evident in the strong respiration pulses after dry soils are re-wetted, known as the Birch effect (Birch, 1958). But studies have shown that a successful simulation of the soil respiration pulses associated to re-wetting events requires the incorporation of additional mechanisms, namely the explicit representation of a bio-available C pool, such as dissolved organic matter (DOC), and a moisture regulation of decomposer's access to this pool that may differ from the moisture regulation on the decomposition reaction itself (Lawrence et al., 2009; Zhang et al., 2014).

The response of soil respiration to temperature and moisture is highly dynamic, both spatially and temporally (Hamdi et al., 2013; Moyano et al., 2012). Moisture and temperature interactions have been observed in a number of experimental studies (Craine and Gelderman, 2011; Rey et al., 2005; Suseela et al., 2012; Wickland and Neff, 2008), but neither consistent trends nor general explanatory theories have been identified. Improving our understanding of these interactions is a crucial step in increasing confidence in models and for interpreting modelling and experimental results (Crowther et al., 2016; Tang and Riley, 2014). Identifying the model structures and parameterizations that can best represent these interactive effects has been attempted by very few studies (Sierra et al., 2015, 2017).



The objectives of this study are, first, to compare the ability of different soil C modelling approaches to reproduce temperature and moisture interactive effects on soil carbon fluxes, and second, to gain insight into the underlying mechanisms from the model comparison. With the hypothesis that a more mechanistic model will be better capable of simulating such interactions, we compare variations of a model based on a microbial model with an explicit representation of a dissolved C pool. We tested

first order, second order, and Michaelis-Menten reaction kinetics in combination with an explicit simulation of diffusion fluxes, and then compare the best diffusion model with versions based on common empirical moisture relationships.

## 2.        Observational data

Measurements of the interaction effects of temperature and moisture on soil respiration fluxes were obtained by incubating a crop field soil at several fixed levels of soil moisture and variable levels of temperature over a period of ca. 6 months, as

detailed in the following.

Soils from 0-20 cm depth were sampled at Versailles, France, from the 'Le Closeaux' experimental field plot, cultivated with wheat until 1992 and with maize since 1993. Mean annual temperature and rainfall are 10°C and 640 mm. The soil is classified as Luvisol (FAO) silt loam (26 % sand, 59 % silt, 15 % clay) containing no carbonates. Organic carbon contents at the start of the incubation were 1.2 % in weight. Soil samples were prepared for elemental analysis (C, N) using a planetary ball mill (3

15    min at 500 rpm). C concentrations were measured using a CHN auto-analyzer (NA 1500, Carlo Erba).

Sampled soils were thoroughly mixed, sieved at 2 mm and stored moist at 4 °C in plastic bags with holes for aeration for 10 days. Soils were then put in small plastic cylinders containing the equivalent of 90g dry soil. To ensure a high and equal water conductivity, all samples were compacted to a bulk density of 1.8 g cm$^{-3}$. The resulting soil porosity was 0.45.

All samples were brought to a pF of 4.2 corresponding to about 7 % mass basis moisture. Three replicate samples were then

adjusted to each of the moisture levels 1, 3, 7, 9, 11, 13, 15, 17, 19, 21, 25 % by weight by adding water or air drying. These values range from air-dry to saturation, with saturation reached at 25 %. Immediately after, the plastic cylinders were put in 500 ml jars containing a small amount of water on the bottom (except for the 1 and 3 % moisture) to prevent soil drying, and equipped with a lid and a rubber septum for gas sampling. Because of the extremely low respiration rates, samples with 1 % moisture were placed in 125 ml jars containing 170 g of soil.

To minimize post-disturbance effects, samples were pre-incubated at 4 °C during 10 days. The samples were then cycled through incubation temperatures following the sequence 5-20-35-5-20-35 °C, thus applying two temperature cycles to each sample. This was done in order to capture possible hysteresis of temperature effects and to reduces the covariance between a temperature response and substrate depletion (helping constrain model parameters). Samples were flushed with $CO_2$ free air and left to accumulate $CO_2$ for 3 to 74 days. The amount of days was chosen so that sufficient $CO_2$ accumulated for the micro

gas chromatographer measurements (at least 100 ppm), thus depending on the soil temperature and moisture content. An air sample was then taken from each soil sample headspace and respiration rates calculated as the accumulated amount over the



accumulation time. This process was performed repeatedly over successive temperature steps over a total incubation period of ca. 6 months (Figure 1).

As shown in Figure 1, the timing of temperature treatments was not equal for all samples, with some temperature steps missing at low moisture levels. This was partly due to the time required for $CO_2$ concentrations in the flask headspace to reach

detectable limits, the time necessary for carrying out measurements and human error. However, while important for a statistical comparison between treatments, such differences are of little consequence when looking at model performance and the fit between model and data, which constitute the results presented here.

## 3.    Modelling approach

### 3.1.    Structure and state variables

We started with a basic soil C model with the following state variables: a bio-unavailable particulate C pool ($C_P$), a bio-available dissolved C pool ($C_D$), a microbial C pool ($C_M$) and two extracellular enzyme C pools, one representing the enzyme fraction at the decomposition site ($C_{ED}$) and one the fraction at the microbial site ($C_{EM}$). With this model we assume two conceptual soil spaces that are separated by a diffusion barrier, one being the site of decomposition and the other the site of microbial uptake and enzyme production (Figure 2). This model thus closely follows Manzoni et al. (2016), and otherwise

builds on other published microbial models (Allison et al., 2010; Schimel and Weintraub, 2003). We refer to those studies for general assumptions and application of this type of model. Aspects specific to this study are described below.

The rates of change of the model state variable were defined as:

$$\frac{dC_P}{dt} = F_{L_SP} + F_{MP} - F_{PD} \tag{1}$$

$$\frac{dC_D}{dt} = F_{L_MD} + F_{PD} + F_{E_DD} + F_{E_MD} - F_{DM} - F_{DR_G} - F_{DE_M} \tag{2}$$

$$\frac{dC_M}{dt} = F_{DM} - F_{MP} - F_{MR_M} \tag{3}$$

$$\frac{dC_{ED}}{dt} = F_{E_ME_D} - F_{E_DD} \tag{4}$$

$$\frac{dC_{EM}}{dt} = F_{DE_M} - F_{E_ME_D} - F_{E_MD} \tag{5}$$

where $F$ represents the flux of C from one pool to another as indicated by the subscripts, so that $F_{PD}$ is the flux from the particulate pool to the dissolved pool. The subscripts $L_S$ and $L_M$ denote input of structural and metabolic litter (as defined by

Parton et al., 1987), which for simulating the incubated soils were set to zero, and $R_M$ and $R_G$ are microbial growth and maintenance respiration.



### 3.2.    Decomposition and microbial uptake

The flux of $C_P$ to $C_D$, $F_{PD}$, represents decomposition of organic matter, a process that in soils is largely driven by the activity of microorganisms. The latter produce exo-enzymes that catalyse the decomposition reaction. $U_D$, represents the total uptake flux by microbes of the water soluble decomposed pool $C_D$ (microbes being the reaction "catalysers"). Conventional soil C

models simulate decomposition as a first order decay reaction. However, more realistic models can be built by using either simple second order or Michaelis-Menten reaction kinetics. Thus, optional ways of modelling both $F_{PD}$ and $U_D$ include:

$$F = V[R] \qquad (6)$$

$$F = V[R][C] \qquad (7)$$

$$F = \frac{V[R][C]}{K + [R]} \qquad (8)$$

where $F$ is the flux, $V$ is a base reaction rate, $K$ is the half saturation constant, R the reactant and C the catalyst. In the case of decomposition, the respective parameters are $V_D$ and $K_D$, and the terms [R] and [C] are concentrations of $C_P$ and $C_{ED}$. In the case of uptake, these are respectively $V_U$, $K_U$, $C_D$ and $C_M$. The three approaches for reaction kinetics were tested in order to

find the best fit between model and data, as described in Sect. 4.

### 3.3.    Diffusive fluxes

Diffusion fluxes depend on a concentration difference, a diffusivity term, and the distance over which diffusion occurs (Manzoni et al., 2016). For the purpose of modelling diffusion in soils, values of diffusivity and diffusion distances are required that best average or represent the actual underlying soil complexity. For practical purposes, we combined these two values into

a single calibrated parameter, a conductance ($g_0$), representing the compound effects of diffusivity and distance. This was done because the values of the latter are unconstrained (from lack of information), and their effects are inversely correlated, so simultaneous calibration would lead to a problem of parameter identifiability. The moisture-scaled conductance ($g$), which in our model is assumed equal for the $C_D$ and $C_E$ pools, is then given by:

$$g = g_0 d_\theta \qquad (9)$$

where $d_\theta$ is a function of soil volumetric water content (VWC or θ):

$$d_\theta = (\phi - \theta_{th})^m \left(\frac{\theta - \theta_{th}}{\phi - \theta_{th}}\right)^n \qquad (10)$$

where $\phi$ is pore space, and $n$ and $m$ are calibrated parameters (Hamamoto et al., 2010; Manzoni et al., 2016), which are variable and were also calibrated in this study. $\theta_{th}$ is the percolation threshold for solute diffusion (Manzoni and Katul, 2014), which



was here a calibrated parameter. The diffusive flux of enzyme C between the microbial and the decomposition spaces is then calculated as:

$$F_{E_M E_D} = g(C_{EM} - C_{ED}) \qquad (11)$$

Diffusion limitations also affect the amount of the dissolved pool ($C_D$) available for microbial uptake. Instead of dividing $C_D$ into a pool for each space, the conductance, $g$, was used as a multiplier of the base uptake rate, $V_U$ (Eq. (6-8)). This served to reduce the number of model pools and parameters while still retaining a diffusivity limitation on this flux.

### 3.4. Microbial and enzyme dynamics

$U_D$ is split into $F_{DM}$, $F_{DE_M}$ and $F_{DE_M}$, representing the fluxes of $C_D$ going to $C_M$, $R_G$ and $C_{EM}$, respectively. These fluxes are defined as:

$$F_{DM} = U_D f_{ug}(1 - f_{ge}) \qquad (12)$$

$$F_{DR_G} = U_D(1 - f_{ug}) \qquad (13)$$

$$F_{DE_M} = U_D f_{ug} f_{ge} \qquad (14)$$

where $f_{ug}$ represents the fraction of uptake going to growth, otherwise known as microbial growth efficiency or carbon use efficiency, and $f_{ge}$ is the fraction of growth going to enzyme production. $C_M$ goes to either maintenance respiration or the $C_P$ pool according to:

$$F_{MP} = C_M r_{md}(1 - f_{mr}) \qquad (15)$$

$$F_{MR_M} = C_M r_{md} f_{mr} \qquad (16)$$

where $r_{md}$ is the rate of microbial decay and $f_{mr}$ is the fraction of that decay that is lost as respiration. $f_{mr}$ thus determines the amount of maintenance respiration and is here assumed to be constant (but note that $r_{md}$ is temperature dependent). The breakdown of enzymes going to the $C_D$ pool, is determined by the rate of enzyme decay, $r_{ed}$, as:

$$F_{E_D D} = C_{ED} r_{ed} \qquad (17)$$

$$F_{E_M D} = C_{EM} r_{ed} \qquad (18)$$

### 3.5. Temperature effects

Reaction rates ($V_U, V_D, K_U, K_D$ in Eq. (6-8)) and decay rates ($r_{ed}$ and $r_{md}$) are temperature sensitive and calculated from their reference values following an Arrhenius type temperature response:



$$p = p_{ref} \, exp\left(-\frac{E_a}{R}\left(\frac{1}{T} - \frac{1}{T_{ref}}\right)\right) \qquad\qquad (19)$$

where $p$ is the temperature modified value for the respective parameter, $p_{ref}$ the reference value at temperature $T_{ref}$, $T$ temperature in Kelvin, $E_a$ the activation energy and $R$ the universal gas constant. Volumetric water content, $\theta$ (m$^3$ m$^{-3}$) and temperature, T, are model input variables.

Temperature thus affects the rates of decomposition and uptake, the half saturation constant in the Michaelis-Menten equation, as well as the rates of microbial and enzyme decay. Apparent activation energies – describing the observed temperature relationship, both in measurements and model data – were obtained by fitting an Arrhenius equation to the temperature-flux relationship at each level of moisture and separately for 5-20 °C and 20-35 °C. $E_a$ was calculated for measured respiration, modelled respiration ($R_G + R_M$) and modelled decomposition ($F_{PD}$).

## 4.        Model calibration and comparisons

Models were calibrated by optimizing a set of parameters to best fit the measured soil respiration data. For this, the model was run to reproduce each sample treatment, i.e. the applied incubation times and temperatures for each level of moisture (Figure 1). Measured and simulated data from all samples were then combined and the overall root mean square error (RMSE) calculated. Calibrated and non-calibrated parameters are shown in *Table 1*. Equilibrium conditions were not assumed at the start of the experimental procedure. Therefore, initial conditions were obtained by also optimizing the fractions of initial carbon pool sizes ($f_P, f_D, f_M$). Total organic C was set equal to the average measured value.

For parameter optimization we used the Nelder-Mead algorithm, as implemented in the function modFit in package FME of the R programming language (R Development Core Team, 2016; Soetaert and Petzoldt, 2010). We used an error term ('err' argument to FME function modCost) to weight the residuals. The error was calculated as the normalized (0-1) standard deviation of measured values at each combination of temperature and moisture. To avoid an unreasonable weighting of measurements with near zero errors, 0.1 was added to the normalized value.

For a visual inspection of the model-data fits, we plotted both the measured and model relationship between soil respiration vs. moisture, soil respiration vs. temperature, and apparent activation energy ($E_a$) vs. moisture content.

## 4.1.    Comparison of reaction kinetics

Models with alternative reaction kinetics were compared by calibrating versions using all combinations of fluxes $F_{PD}$ and $U_D$ using Eq. (6-8). Thus, we tested all combinations of first order, second order, and Michaelis-Menten kinetics reaction kinetics for both decomposition and uptake. We then evaluated the model-data fit based on RMSE values as well as on a visual inspection of the plotted relationships. A "best" model was then selected for further analysis.





## 4.2.    Comparison of moisture regulations: diffusion versus empirical

A second model comparison was carried out to test the impact of different approaches for modelling moisture effects. For this we modified the model M2-dif (*Table 2*) removing diffusion fluxes and adding empirical moisture functions. This consisted in removing all diffusion effects (so that $C_{EM}$ and $C_{ED}$ were replaced by a single $C_E$ pool and the uptake rate, $V_U$, was no longer modified by $g$) and adding a function to scale (i.e. multiply) the decomposition flux, $F_{PD}$. This approach is equivalent to the conventional way used to model moisture effects on soil C fluxes. Two alternative moisture scaling functions were tested (Moyano et al., 2013), one based on relative water saturation (M2-sat) and the other on water potential (M2-wp):

$$f(\theta_S) = a\theta_S - b\theta_S{}^2 \tag{20}$$

$$f(\Psi) = \max\left\{min\left\{\begin{array}{c} 1 - \dfrac{\left[\log_{10}(\Psi) - \log_{10}(\Psi_{opt})\right]}{\left[\log_{10}(\Psi_{th}) - \log_{10}(\Psi_{opt})\right]} \\ 1 \\ 0 \end{array}\right.\right. \tag{21}$$

were $\theta_S$ is relative water saturation, $\Psi$ is soil water potential and a, b, $\Psi_{opt}$ and $\Psi_{th}$ are fitted parameters. The latter two represent the optimal water potential for decomposition and a percolation threshold water potential (equivalent to $\theta_{th}$ in Eq. (10)), and have values close to -0.03 and -15 MPa, respectively. Water potential was calculated based on Campbell (1974) and Cosby et al. (1984).

## 5.    Model steady state, sensitivity analysis and validation

Equations for steady state were derived by setting the rate of change in the state variables to zero in Eqs. 1-5 (where the flux terms are replaced by their respective equations), and then solving for the state variables. This was performed in Python using the "sympy" package (Meurer et al., 2017).

A sensitivity analysis was carried out on all model parameters. For this we simply used the default "sensFun" function from the R package FME, which perturbs each parameter individually by a small amount. We ran the model as above, i.e. simulating the incubation, and using daily output. Daily sensitivities were then averaged to obtain an overall value. The sensitivity was calculated for the $C_P$ pool alone, as this pool represents the largest fraction of soil C.

For model validation, we used soil respiration data from the study by Rey et al. (2005) where a Mediterranean oak forest soil was incubated for one month in a full factorial design at 100, 80, 60, 40 and 20 % of water holding capacity and at 30, 20, 10 and 4 °C. This soil differed from the one used for model calibration in at least 3 aspects: the amount of organic C (7 %), soil pore space (65 %), and texture (classified as silty clay loam). The optimized set of parameters from model M2-dif was used with the exception of the initial fraction of C pools ($f_P, f_D, f_M$) and the percolation threshold ($\theta_{th}$), which we chose to calibrated against the new data with the same procedure as above. The former was required since we had no information to estimate the



microbial, dissolved, and enzyme C for this study. In the case of $\theta_{th}$, we assumed that this parameter is especially sensitive to variations in soil texture and structure. Calibration was then necessary as we did not have a formula to derive it for the new soil (although in previous studies it has been determined as equal to a water potential of -15MP, this value did not give good results in our analysis).

## 6.    Results

### 6.1.    Reaction kinetics

Using different reaction kinetics resulted in a strong variation in model performance as measured by RMSE (*Table 2*). Changes in RMSE were most sensitive to the kinetics of decomposition ($F_{PD}$), with models using Michaelis-Menten kinetics resulting in distinctly lower RMSE values compared to 1st and 2nd order kinetics. On the other hand, different reaction kinetics for the

10 uptake flux, $U_D$, had a much smaller impact on the RMSE, being slightly lower for 1st order kinetics.

Given the small difference in performance between different uptake reaction kinetics, we chose to work with 2nd order kinetics for further analysis (model M2-dif, $R^2 = 0.84$, Figure 3), as this is a closer representation of the underlying mechanisms actually driving this flux, i.e. uptake does not occur without microbial mediation. It also requires less parameters than Michaelis-Menten and is thus a compromise in complexity. (We note that the choice between 1st and 2nd order uptake had a small impact on this

study's results. On the other hand, Michaelis-Menten uptake kinetics had the poorest agreement with observations when comparing the plotted relationships. Plots for the three models based on Michaelis-Menten decomposition can be found in the Supplement material, Fig S1 and S2) The decomposition and uptake equations of the model M2-dif are thus:

$$F_{PD} = V_D C_{ED} C_P / (K_D + C_P) \qquad (22)$$

$$U_D = C_D C_M V_U g \qquad (23)$$

### 6.2.    Moisture regulation

Replacing diffusion effects with empirical moisture scalars, followed by re-calibration, decreased model performance

compared to a diffusion based model, both when using relative water saturation (M2-sat) and water potential (M2-wp) functions (*Table 2*). Although it was possible to simulate the shape of the respiration-moisture relationship for a specific temperature, empirical functions were unable to capture the variation of this response across temperatures, as seen in the measurements and best simulated by the diffusion base model (Figure 4). The diffusion based model more accurately simulated a linear relationship between respiration and moisture at lower temperatures and a steep increase followed by a plateau at high

temperatures. An intermediate response was seen at 20 °C.





### 6.3. Model steady state, sensitivity analysis and validation

Model steady state equations are provided in the Supplement material. For 20 °C, 30 % VWC, 1.2 g $d^{-1}$ C input, and 30 cm soil depth (z), the equilibrium sizes of the model pools are: 2800, 50, 7 and 0.4 g C for the $C_P$, $C_D$, $C_M$ and $C_{ED}$ pools respectively. These values are stable over most of the moisture range and increase exponentially only at very low soil moisture

(data not shown). A similar pattern was observed for temperature, with the $C_P$ pool increasing towards high values only at temperatures near 0 °C. The same pool showed little sensitivity to changes in C input.

Table 1 shows the averaged values from the sensitivity analysis done on the model $C_P$ pool. The highest sensitivities were found for parameters that affect the diffusion fluxes, with the $n$ exponent in Eq. (10) having the largest effect, followed by the base conductance, $g_0$. Large effects were also seen for most activation energy parameters, denoting a strong general effect of

10 temperature. Also high were the sensitivities to $K_D$ and $f_{ug}$, reflecting the importance of Michaelis-Menten kinetics for decomposition and carbon use efficiently, respectively. Notably low sensitivities were found for rates of microbial and enzyme decay.

Simulation of the incubated soil from the study of Rey et al. (2005) resulted in a very high fit to the validation data, with an RMSE of 0.09 in fluxes that were almost an order of magnitude higher than those used for calibration, and a model $R^2$ of 0.99

(Figure 6). This was reflected in a generally good agreement between the relationships of model and observations with moisture (Figure 7) and temperature (Figure 8).

### 7. Discussion

The interaction often observed in the effects of temperature and moisture on the cycling of soil C is an indicator of the complex nature of soil systems. Such responses are often ignored, particularly by modellers trying to minimize model complexity and

20 derive functions that are easy to parameterize, but also by experimentalists focusing on finding an invariable response to a single factor. But a careful consideration of the nature of soils suggests that interactions should be expected, something that becomes evident in multi-factorial experiments as well as in field measurements. Here we found clear interactive effects in our experimental observations, adding to the evidence that fixed empirical temperature and moisture scalars, as used in conventional soil C models, are inappropriate for simulating the variability often found in natural conditions.

When fitting models to the data, we were unable to attain a close fit when using first or second order reactions kinetics for decomposition. In fact, the resulting $R^2$ values were highly negative, meaning that the models were worse predictors than the simple mean of the data. Since the total amount of soil C in our samples was equal among samples and its relative change in the six months of incubation was small, we expected that simple second order kinetics would do as well as Michaelis-Menten, assuming that the parameter values are adjusted to compensate for the different equation forms. The poor fits we saw suggested

that the optimization algorithm remained in a local minima of the cost function. However, better fits were not attained with $2^{nd}$ order kinetics even after starting optimization with a higher initial $V_{D\_ref}$ to compensate for the Michaelis-Menten effects of $K_D$. This, combined with the fact that the model was more than twice as sensitive to a change in $K_D$ compared to $V_D$, would





indicate that Michaelis-Menten kinetics are in fact important for explaining soil C flows. Indeed, even in this case where the $C_P$ pool is relatively invariant, the outcome of a strong temperature effect modifying $K_D$ ($E_a$ of 89kJ) cannot be reproduced by simple 2nd order kinetics.

The relative importance of different processes was also shown by the model parameter sensitivity values. It is perhaps not

surprising that the highest values were related to diffusion and temperature, since these were the two factors that varied in our experiment. However, these factors also vary considerably in natural ecosystems, so the values remain informative. The high sensitivity found for $f_{ug}$ also demonstrated the importance of C use efficiency of microbes, with the optimized value of 0.7 coinciding with that obtained by Haggerty et al. (2014).

Since optimizing all parameters against our data resulted in an $R^2$ of 0.84, it was surprising that model validation gave an $R^2$

of 0.99. We note that few studies were found with data on moisture and temperature interactions and this was the only validation attempt carried out. The very high $R^2$ is largely thanks to a recalibration of initial pool sizes and probably also to the simpler experimental design compared to our study. There were only 20 data points in the validation data, one for each temperature and moisture combination. With 3 replicates, 11 moisture levels and 2 temperature cycles, we had more data and more variability. Despite the above and this being just a first validation step, such a close agreement using independent data

and a soil that differed considerably in C content, provides strong support to the model.

Model steady state or equilibrium is attained when the rate of change of all state variables equals zero, reflecting the state towards which the system will tend under invariant input and forcing conditions. A steady state is never attained in natural systems, where external drivers are in constant change, but they help evaluate how the model behave under specific conditions. Results here showed that the model gives realistic values in the range of temperature for which it was calibrated, but leads to

unrealistic values under colder conditions. In addition, the $C_P$ pool shows little sensitivity to changes in C input. Clearly, while the model fitted well the validation data, it should not be extrapolated outside the used ranges and should not be applied for field simulations. The limitations encountered are characteristic of non-linear microbial models and mark their limitations as predictive tools. However, such limitations are most likely the result of missing processes that still need to be adequately represented. Recent work has shown, e.g., that a density dependent mortality rate of the microbial pool can lead to much more

realistic long term simulations (Georgiou et al., 2017). Leaching of $C_D$ is another example that could significantly affect C dynamics. Such mechanisms where not essential for simulating our observations but will need to be assimilated for extending the application of these models.

Other limitations for simulating soil C cycling using this type of model can be pointed out. Our results cannot be extended to litter decomposition (Cotrufo et al., 2015) or organic soils, which will be much more dependent on substrate quality and less

affected by carbon diffusion (Manzoni et al., 2012). Also, oxygen supply, which is critical in peatlands and other soils (Clymo, 1984; Frolking et al., 2001), was not taken into account. Finally, we did not include mineral adsorption of carbon as an active mechanism in this study. This is contrary to recent studies that used adsorption-desorption fluxes to explain the variability in temperature responses (Tang and Riley, 2014). However, values of mineral desorption rates found in the literature (Ahrens et





al., 2015) suggest that these rates, although important in the long term, are too slow to have a significant effect on the time scale of this or similar experiments, and thus on most estimates of soil respiration temperature sensitivities.

### 7.1. Temperature effects

Unlike other calibrated parameters, the activation energy values for microbial ($E_{a\_m}$) and enzyme ($E_{a\_e}$) decay were fixed at 10 kJ, representing a positive but low temperature sensitivity. This value was used in order to be consistent with two main observations:

a) *The effect of $E_{a\_m}$ on the amount of microbial carbon.* A high $E_{a\_m}$ results in large changes of microbial biomass C with temperature. However, observations often show a negative but moderate effect of temperature on microbial biomass (Grisi et al., 1998; Salazar-Villegas et al., 2016).

b) *The effect of $E_{a\_e}$ on carbon decomposition rates.* High $E_{a\_e}$ values result in increasing accumulations of soil C with warming (Allison et al., 2010; Tang and Riley, 2014) as a consequence of a decrease in the enzyme pool caused by accelerated turnover. This is a critical aspect of enzyme driven soil carbon models and largely determines simulated responses to long term warming. Experimental evidence for $E_{a\_e}$ is lacking, but the latest observations of mid-term responses to warming are compatible with low values (Crowther et al., 2016).

All optimized $E_a$ values came close to 90kJ, which translates to a fairly high $Q_{10}$ range of 3-4. Interestingly, however, the actual relationship with temperature of both the observed and modelled $CO_2$ production indicated a much lower sensitivity, approximating the commonly measured $Q_{10}$ value of 2. These values were mostly stable at high levels of soil moisture, but increased sharply under drier conditions.

On the other hand, the relationship we observed and were able to simulate seems by no means to be general, but rather to depend strongly on the system's initial conditions. This became evident in the validation step, where the apparent temperature sensitivities, both in the observations and the model, remained close to an $E_a$ of 90 kJ and thus much closer to the parameterized $E_a$ values. Also the change in $E_a$ with moisture content followed a different trend in the validation data, although again values increased with lower moisture. We thus could see that, with the same underlying temperature sensitivities but different soil pool sizes (and possibly different diffusion limitations), different conclusions on the temperature sensitivity of soil C decomposition can be reached when looking only at measured respiration fluxes.

Decomposition, which was only modelled, consistently showed a lower apparent temperature sensitivity than respiration, with a $Q_{10}$ between 1-2 for our experiment and just below 3 for the validation study. Arguably, these values are the most relevant for predicting long term changes, since uptake and respiration ultimately depend on C made available by decomposition. Why these values remained especially low and how they may change in the long term remains to be explored, but rather low sensitivities are consistent with some integrative studies at the ecosystem level (Mahecha et al., 2010). Such results raise the question of what $E_a$ or $Q_{10}$ values – i.e. the apparent for respiration, apparent for decomposition, or the parameterized – are adequate when applying conventional empirical soil models. Since these model will tend to have similar apparent and intrinsic




behaviour, the answer is not clear and will require further research. Ultimately, the better option may be to abandon such model and develop better validated mechanistic alternatives for prediction purposes.

## 7.2.     Moisture effects and diffusion limitations

Diffusion fluxes are a function of water content, diffusivity coefficients and pool concentrations. Different equations have

been used to calculate diffusion as a function of water content in soils (Hamamoto et al., 2010; Hu and Wang, 2003). All these equations generally predict a strong positive near exponential effect of water content on diffusion. Following previous studies (Manzoni et al., 2016), we chose the function from Hamamoto et al. (2010). This equation allows for an adjustment of the percolation threshold ($\theta_{th}$) in different soils. We note that when using the $\theta_{th}$ obtained during calibration (0.063) we also obtained a high fit to the validation data ($R^2 = 0.98$, data not shown), so the recalibration of $\theta_{th}$ led to a noticeable but small

improvement. While the value 0.063 for our soil came close to the water potential of -15 MPa found in previous studies (Manzoni and Katul, 2014), this relationship did not hold for the validation soil, where we assumed a higher clay and silt content from its classification. Thus, a prerequisite for applying our model to other soils is finding a relationship between $\theta_{th}$ and soil type that holds in all cases.

Diffusion regulations can be implemented either by simulating two separate pools between which diffusion takes place or by

determining the available amount of a pool as a function of diffusivity (or conductance in our case) at each time step. In our model we used a combination, simulating a diffusion flux between enzyme pools and calculating the how much $C_D$ is available for uptake at each time step. We did not assume a diffusion regulation of available particulate C, an approach that is closer to empirical functions scaling the decomposition flux directly and that has been implemented in other microbial models (Davidson et al., 2012).

In our study especially, but also in the validation data, the moisture response tended to become less linear and have a larger plateau at higher temperatures. The mechanisms leading to such interactions are still unclear, but our model comparison indicates that diffusion limitations play a central role. The plateau behaviour, a decrease near saturation, and even near linear responses, all contrast with the near exponential relationship between moisture and conductance given by Eq. (10) and with the fact that no oxygen limitations at high moisture were modelled. They may, however, result from a faster depletion of

available carbon at high moisture and at high temperatures, driving down the accumulated fluxes over time.

In models where decomposition and respiration are separated processes, these fluxes can show different responses. This decoupling is especially evident when diffusion limitations come into play. Plots of modelled fluxes against temperature and moisture (Fig. S3) showed a different relationship when comparing respiration and decomposition. Figure 9 shows modelled decomposition against respiration (using M2-dif) as accumulated values, each line being a sample at a different water content.

Without any diffusion limitation, the relationship follows a slope of ca. 0.3, determined by $1-f_{ug}$, where $f_{ug}$ is the fraction of uptake going to growth (the C use efficiency). This slope, however, changes as diffusion becomes limiting, and temperature seems also to play a role as evidenced by the shifts in the slope occurring at various intervals. With time these fluxes will tend to equilibrate as the $C_D$ and $C_{ED}$ pools adjust. But the proportionality between these fluxes is not constant in will depend on



moisture, temperature, and time, even after months of incubation. These results show that, without a proper modelling framework and when assuming a constant proportionality, interpretations based only on respiration activity may lead to wrong conclusions about the dynamics of organic matter decomposition, especially at low moisture contents and in short and mid-term experiments.

# 5   Conclusions

As the main mechanism linking water content with the movement of substrates, microbes and enzymes, diffusion plays a central role in soil organic matter decomposition. We here showed that integrating it into models can significantly improve our understanding of soil C dynamics. Not only was our diffusion-based model better at simulating the effects of moisture variability, it also improved the simulated temperature responses, thus allowing for a better interpretation of the observed temperature sensitivities. This and similar studies indicate that measured temperature sensitivities cannot be generalized or correctly interpreted without having a full understanding of the relevant soil factors involved, their interactions, and the state of soil carbon and microbial pools.

Creating models that capture the variability in the response of C dynamics across different soils and at different levels of driving factors remains challenging. However, process based models are of central importance for establishing confidence in C cycle predictions and soil-climate feedbacks. As seen here, the structure and process representation of models can be critical for simulating the complex response of soil C fluxes to combined changes in temperature and moisture. Diffusion as a moisture regulation of soil C fluxes has not been used in large scale predictions, which still rely on empirical scaling functions. Evidence of interactions seen in experiments and presented here from a mechanistic model perspective indicate that these simpler approaches do not always hold. Further research should focus on more extensively validating these approaches and finding the relationships necessary for extending the application of such models to diverse soil types.

# Code availability

All code used for this analysis is available at https://doi.org/10.5281/zenodo.1208756

# Data availability

Data used for this analysis is available at https://doi.org/10.5281/zenodo.1208756

# Author contribution

F.M. developed the model, analysed the data and wrote the paper. N.V. carried out the lab experiments and revised the paper. L.M. contributed to model optimization, discussions and revisions.



**Competing interests**

The authors declare no competing interests.

**Acknowledgements**

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



**Figures**

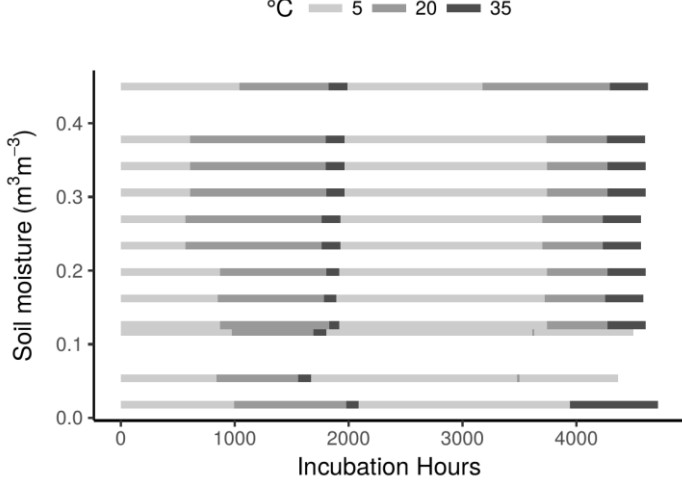

**Figure 1: Graphical representation of the incubated soil samples showing the fixed levels of moisture content and the times at different temperatures.**

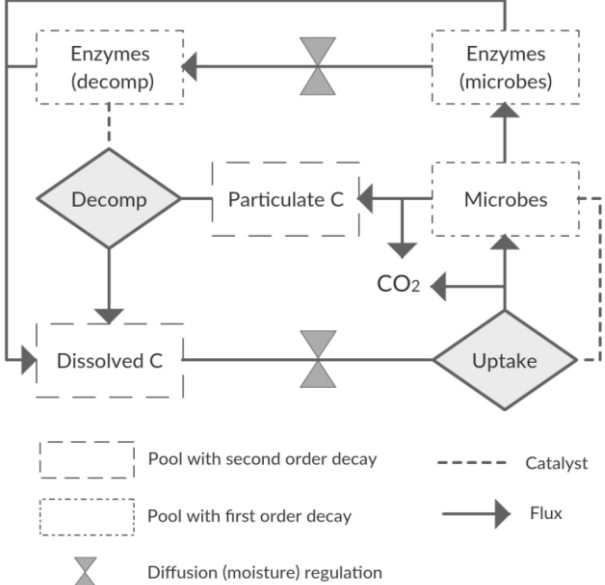

**Figure 2: Diagram showing C pools and fluxes, as well as the points of diffusion limitations. Second order decay may refer also to Michaelis-Menten reaction kinetics. Variations of this scheme were tested in this study.**





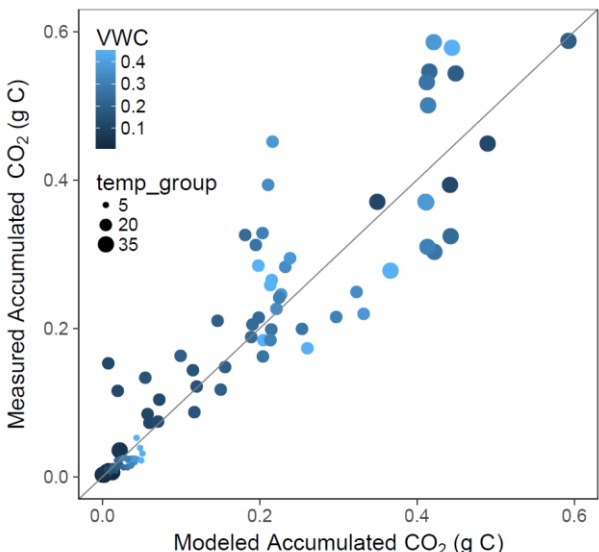

**Figure 3: Model vs measured accumulated $CO_2$ of incubated soil samples. Colour depicts the range of volumetric water content (VWC) and size the temperature group. The model $R^2$ is 0.84.**



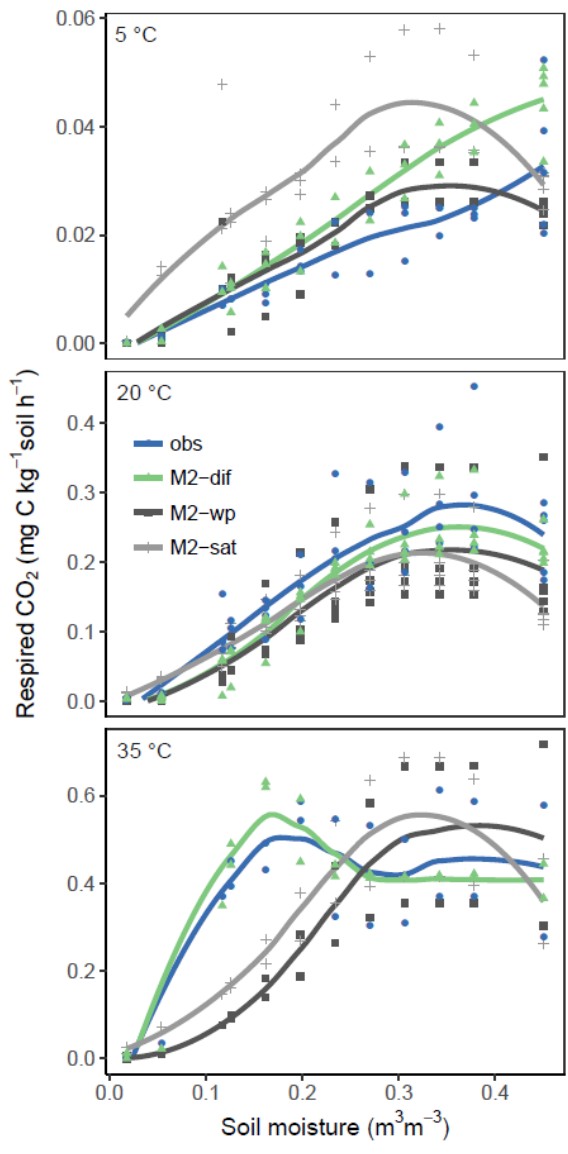

**Figure 4: Relationship of soil respiration with volumetric soil moisture. Results shown over three temperatures levels (5, 20, 35 °C) for the observed data (obs) and three model versions (M2-dif, M2-wp and M2-sat). Broad lines are smooth loess fits depicting the mean relationship.**





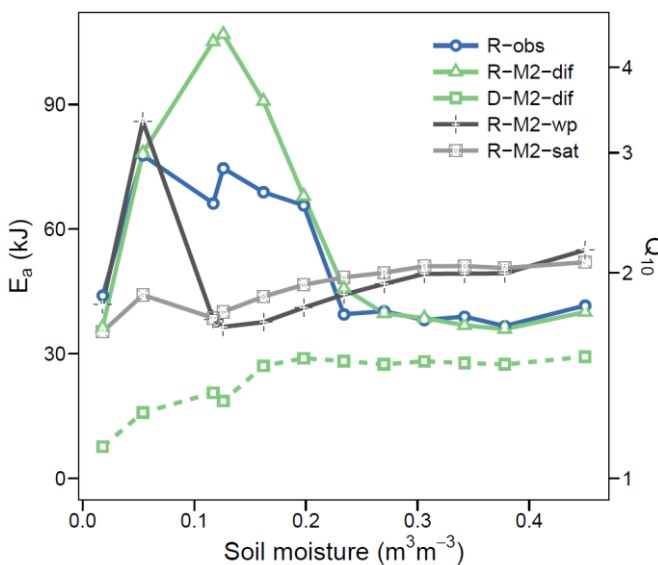

**Figure 5: Fitted temperature sensitivities of respiration and decomposition fluxes, showing activation energy ($E_a$) fitted using the whole temperature range (5-35 °C) and the equivalent $Q_{10}$ derived for the temperature range 15-25 °C. Sensitivities are shown for respiration fluxes of observational data (R-obs) and of three model versions (R-M2-dif, R-M2-wp and R-M2-sat). For comparison, the sensitivity of the decomposition flux from model M2-dif is also included (D-M2-dif).**

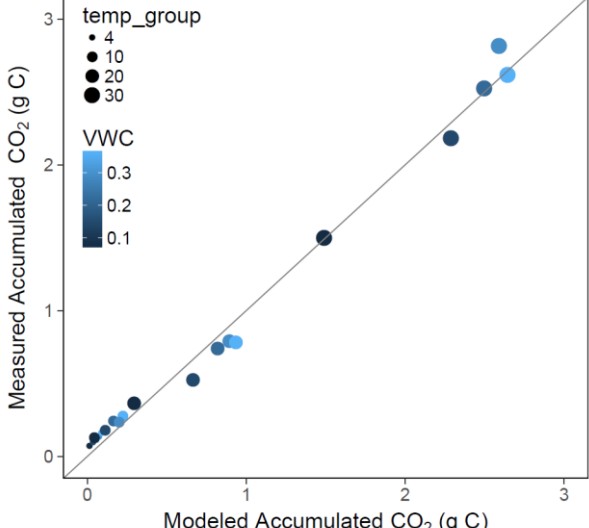

**Figure 6: Model vs measured accumulated $CO_2$ after simulating the experiment from Ray et al. (2005). Colour depicts the range of volumetric water content (VWC) size the temperature group. The model $R^2$ is 0.99.**





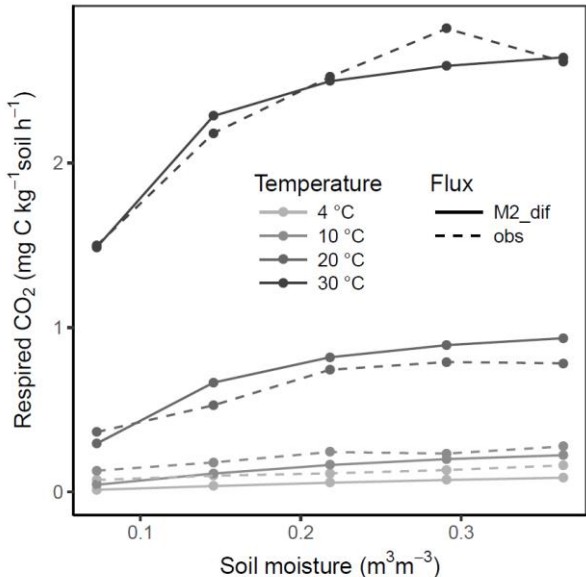

**Figure 7: Relationship of soil respiration with volumetric soil moisture shown for model M2-dif and observations from the validation data (obs). Results are shown over four temperatures levels.**

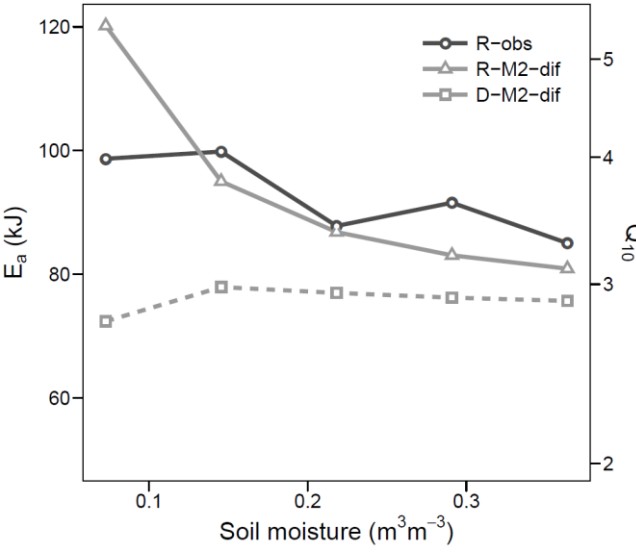

**Figure 8: Fitted temperature sensitivities of respiration and decomposition fluxes, showing activation energy ($E_a$) fitted using the whole temperature range (5-35 °C) and the derived equivalent $Q_{10}$ valid for the temperature range 15-25 °C. Sensitivities are shown for respiration fluxes of the validation data (R-obs) and for the respiration and decomposition flux of model M2-dif.**

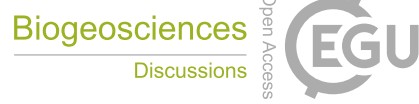



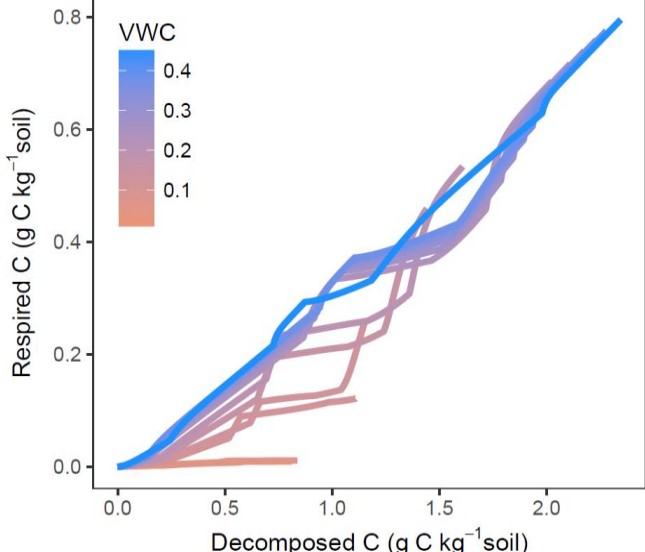

**Figure 9: Modelled decomposed vs respired C shown as accumulated values over the entire simulated incubation. Each line is a sample at a different moisture content.**




**Tables**

*Table 1: Model parameters, calibrated and non-calibrated. Calibrated parameters are rounded to two significant digits. The Sens column values are a relative measure of the sensitivity of the model $C_P$ pool to small perturbations in the parameter values.*

| Name | Value | Units | Sens | Description | References |
|---|---|---|---|---|---|
| | | | | *Calibrated parameters* | |
| $g_0$ | 2.2 | h$^{-1}$ | 4.6 | Conductance for diffusion | (Hu and Wang, 2003; Jones et al., 2005; Manzoni et al., 2016; Vetter et al., 1998) |
| $E_{a\_K}$ | 89 | kJ | 4 | Activation energy for $K_D$ | - |
| $E_{a\_r}$ | 95 | kJ | 3.8 | Activation energy for $r_{mr\_ref}$ | (Price and Sowers, 2004) |
| $E_{a\_V}$ | 87 | kJ | 3.6 | Activation energy for $V_{D\_ref}$, $V_{U\_ref}$ | (Tang and Riley, 2014; Wang et al., 2013) |
| $f_D$ | 9.2e-5 | - | 0 | Initial $C_D$ fraction of SOC | - |
| $f_E$ | 6.1e-4 | - | 0.04 | Initial $C_{EM}$ / $C_{ED}$ fraction of SOC | - |
| $f_M$ | 0.071 | - | 0.45 | Initial $C_M$ fraction of SOC | - |
| $f_{ug}$ | 0.7 | - | 2.5 | Fraction of $U_D$ going to growth | (Hagerty et al., 2014) |
| $f_{ge}$ | 0.025 | - | 0.04 | Fraction of growth going to $C_{EM}$ | (Schimel and Weintraub, 2003) |
| $K_{D\_ref}$ | 50 | kg C m$^{-3}$ | 4.1 | Michaelis-Menten constant | - |
| $n$ | 2.3 | - | 6.4 | Exponent in Eq. (10) | (Hamamoto et al., 2010) |
| $m$ | 1.2 | - | 0 | Exponent in Eq. (10) | (Hamamoto et al., 2010) |
| $r_{ed\_ref}$ | 5e-4 | h$^{-1}$ | 0.03 | Reference rate of $C_{EM}/C_{ED}$ decay | (Allison et al., 2010; Li et al., 2014) |
| $r_{md\_ref}$ | 1.5e-3 | h$^{-1}$ | 0.03 | Reference rate of $C_M$ decay | (Allison et al., 2010; Li et al., 2014) |
| $r_{mr\_ref}$ | 4.2e-5 | h$^{-1}$ | 0.001 | Rate of maintenance respiration | (Price and Sowers, 2004) |
| $V_{D\_ref}$ | 0.35 | h$^{-1}$ | 1.5 | Reference rate of decomposition | (Allison et al., 2010; Li et al., 2014) |
| $V_{U\_ref}$ | 0.092 | h$^{-1}$ | 0.35 | Reference rate of carbon uptake | (Allison et al., 2010; Li et al., 2014) |
| $\theta_{th}$ | 0.063 | m$^3$m$^{-3}$ | 0 | Moisture threshold for diffusion | (Manzoni and Katul, 2014) |
| | | | | *Non-calibrated parameters* | |
| $T_{ref}$ | 293 | °K | - | Reference temperature | - |
| $E_{a\_m}$ | 10 | kJ | 4.6 | Activation energy for $r_{md\_ref}$ | (Grisi et al., 1998; Salazar-Villegas et al., 2016) |
| $E_{a\_e}$ | 10 | kJ | 2.2 | Activation energy for $r_{mr\_ref}$ | (Grisi et al., 1998; Salazar-Villegas et al., 2016) |



***Table 2: Different model versions with their weighted and unweighted root mean squared errors (RMSE) and $R^2$ after parameter calibration (in units: mgC kgSoil$^{-1}$ h$^{-1}$). $F_{PD}$ = decomposition flux, $U_D$ = dissolved C uptake flux, 1$^{st}$ = first order kinetics, 2$^{nd}$ = simple second order kinetics, M = Michaelis-Menten kinetics. The remaining combinations of 1$^{st}$ and 2$^{nd}$ order reaction kinetics showed similarly high RMSE and are not shown.***

| Model name | $F_{PD}$ | $U_D$ | Moisture effect | RMSE (weighted) | RMSE (unweighted) | $R^2$ |
|---|---|---|---|---|---|---|
| 11-dif | 1$^{st}$ | 1$^{st}$ | Diffusion | 4.34 | 1.08 | -42 |
| 22-dif | 2$^{nd}$ | 2$^{nd}$ | Diffusion | 4.45 | 1.78 | -44 |
| M1-dif | M | 1$^{st}$ | Diffusion | 0.21 | 0.06 | 0.86 |
| M2-dif | M | 2$^{nd}$ | Diffusion | 0.24 | 0.07 | 0.84 |
| MM-dif | M | M | Diffusion | 0.24 | 0.08 | 0.82 |
| M2-sat | M | 2$^{nd}$ | Eq. (20): | 0.29 | 0.09 | 0.71 |
| M2-wp | M | 2$^{nd}$ | Eq. (21): | 0.33 | 0.11 | 0.59 |