# Peer review of "Diffusion limitations and Michaelis-Menten kinetics as drivers of combined temperature and moisture effects on carbon fluxes of mineral soils"

_Biogeosciences, 2018_

## Referee Comment (RC1) · R. Grant (Referee) · 16 Apr 2018

General comments This is an interesting model study that makes a key point that higher order kinetics are needed to model respiration responses to changes in temperature and soil water content. This point is important because many SOC models still retain first order kinetics when making projections of climate change effects on SOC and hence on climate feedbacks, with a possible risk of error. Of particular interest to me but not often considered in modelling was the reduced sensitivity to temperature of microbial decay vs that of uptake and growth, and the point raised in the Discussion about density-dependent microbial decay, as both these processes affect microbial biomass

and hence decomposition rates in higher order models. The authors have done a good job in testing model performance against experimental results from soil incubations. However a more constrained testing of its performance would be achieved by comparing the actual time courses of CO2 emissions measured during the incubations, rather than just the totals measured over the duration of each treatment as done in this study. Two key areas need to be developed for this model to be capable of more robust performance and hence wider application: the coupling of C with N for all transformations, because kinetics of decomposition and respiration are strongly affected by SOC quality, and the simulation of O2 limitations on microbial activity rather than fitting declines in activity with higher soil water contents. Both these areas are already well developed in some ecosystem models. Specific comments I would like to see several points about the model addressed before acceptance for publication, as indicated below: Abstract p.2. l.15: a strong effects ? Introduction p.2 l.29: But note more rapid soil N mineralization, uptake, NPP and litterfall that may offset this feedback. N cycling very much needs to be included in any soil SOC model and the authors need to acknowledge this. p.4 l.16: Soil grinding and mixing will increase microbial access to SOC from that in a natural soil, likely raising decomposition rates. p.4 l.18: A BD of 1.8 is inconsistent with a porosity of 0.45. One or the other must be checked. p.4 l.21: The saturated value of 0.25 is less than the porosity of 0.45. p.5 l.6: This statement is valid as long as the model simulates experimental protocol (e.g. duration of treatments).

Modelling approach p.7 l.3: Although if diffusion limitations reduce FPD you will also reduce CD and hence uptake, so make sure there isn't a duplicated effect caused by direct diffusion limitation to CD. p.7 l.7: Check variable names. p.7 l.13: But temperature sensitivity of fmr is different from that of growth. p.8. eq.19: Low and high temperature inactivation terms are often used with Arrhenuis equations to give greater Q10 at low temperature and must lower Q10 at higher while using biologically realistic values of Ea (typically ca 65 kJ mol-1).

Model Calibration Was a spinup run used to enable key state variables to stabilize at

values independent of those initialized? This is standard modelling protocol. p.8 l.25: This is a commendable objective because some SOC models still retain first order algorithms. p.9 eq. 20, 21: Reductions of f (theta) and f(psi) at higher theta and psi are caused by O2 deficiency as noted later in the text, and are better modelled as such because these reductions are temperature-dependent. p.10 l.4: MPa

Results p.11 l.6: How were C inputs evaluated, as in natural ecosystems these also vary with temperature and swc. In fact, these inputs are the most important part of a SOC model as they are the main drivers of microbial activity. p.11 l.13-14. This is a nice test of the model. Describe how values for initialization of C pools and threshold swc were determined for this study. How did these values affect model results, particularly without model spinup? Ideally you should just change total SOC as determined from the soil measurement, and develop rules for allocating total SOC to initial C pools depending on site conditions, and then spinning up the model to equilibrium before comparison with observed values. An even better test of the model would be against the actual time course of CO2 effluxes measured during each incubation, as has been done in earlier modelling studies (e.g. Soil Sci. Soc. Amer. J. 58:1681-1690). This test lets you see whether the model is really simulating the temporal dynamics of respiration at different water contents under changing temperatures.

Discussion p.12 l.10. Specify these changes as noted in Results to establish how robust the model really is. p.12 l.19-20. Would this problem be addressed by a cold temperature inactivation term in eq. 19?

p.12 l.29-30. The absence of O2 limitations is likely causing the reductions in Ea and Q10 at higher swc in Fig. 5. Modelling these limitations should be a key next step in model development. These limitations are already simulated in some other ecosystem models.

p.13 l.4-5: The reduced temperature sensitivity of microbial and enzyme decay needed to model realistic biomass at different temperatures is an important finding of this study.

p.13 l.15: Experimental determinations of Ea are often in the 65 kJ mol-1 range. The larger value modelled here may have been required in the absence of a cold temperature inactivation term in eq. 19.

p.13 l.16-18. Lower values probably arise from O2 limitations. The authors realistically address the current limitations of the model.

p.13 l.32: models

p.14 l.8-9. Why not make the percolation threshold depend on soil water potential (e.g. -15 MPa)? This might improve model robustness by reducing reparameterization for each soil.

p.14 l.30-21: Would CUE decline at higher temperatures if Rm (fmr in (16)) increased exponentially with temperature, as it is known to do?

.

---

## Referee Comment (RC2) · T. Wutzler (Referee) · 17 Apr 2018

Moyano et al. compare several versions of SOM turnover models with a comprehensive set of observations of varying temperature and soil moisture. They show that explicitly accounting for diffusion, compared to using empirical formulation of the temperature/moisture rate modifiers, improves fit and understanding of SOM decomposition. This result is interesting to the soil model developers and biogeoscientists studying SOM turnover and consequences at soil core to larger scales. The paper contains a strong validation by a good agreement with independent data. The clarity of the discussion on reasons for the good validation fit, interactions with initial pools, and different

resulting temperature sensitivities can be improved.

With an extension of discussion and some more clarifications in the discussion, the paper could be published. Nevertheless, I suggest several additional tasks with this model and data, that would help the community.

General comments:

First, while the paper already contains three different structural versions of decomposition, I suggest including another version of an inverse Michaelis-Menten dynamics for depolymerization (but not for DOM uptake), where the non-linear term is in enzymes instead of the substrate ($F = V\ C\_P * C\_E/(K + C\_E)$). This would broaden the application of conclusions of this study, because the inverse formulation is used by many microbial models since suggested by Schimel and Weintraub 2003.

Second, the study describes a decoupling between depolymerization and microbial uptake at low diffusion rates, I assume by accumulating OM in the dissolved pool. Are the fluxes correlated again for the same treatment, if you aggregate over say two weeks? The decoupling is a challenging fact for upscaling studies, that often assume the DOM pool in quick quasi steady state with decomposition and microbial uptake. For low moisture the decomposed flux was almost not taken up and respired (Fig. 9). Would this also be true with two separate DOM pools after longer time? I would appreciate an extended discussion on this topic.

The model used enzyme pools split to locations but a simplified diffusion limited rate multiplier for DOM. What is the reasoning for this decision, and what are the expected consequences for using a rate modifier for enzymes too?

There is an interesting differentiation between parameterized temperature sensitivity ($E\_a$) and an apparent predicted one, the latter one also depending on partitioning of the pools (P13l20ff). What are the reasons and consequences here. The paper would profit from an extended discussion here.

Specific comments:

p5l10: The choice of the wording "particulate" suggests to OM floating together with the DOM. I assume instead that C_P comprises litter and residues also sitting on surfaces. When using "polymeric" it conveys a different connotation and still the "P" can be used as acronyms.

p7l10: The model assumes enzyme production to be modeled similar to growth respiration as a fraction of uptake, instead similar to maintenance respiration as determined by microbial biomass. What are the reasons for this formulation ?

p8l4ff: The wording here suggests, that all the processes have the same temperature sensitivity, i.e. same E_a. I suggest adding another index to E_a that this parameter varies between processes.

P8l10: I assume there is only one set of parameters fitted to the entire data of all temperature and moisture treatments. Would be nice to state that here. Please, also state the number of fitted parameters, and add the initial partitions to Table 1. The fitted parameter vector in a ∼20 dimension space is quite challenging for a gradient based search. Did you check global convergence by starting from more states, maybe more random distributed as just the one described for p11l30. What are the most important correlations in parameter estimates?

P8l10: model calibration open questions: How were enzyme pools initialized? What were the values of fractions for particulate, dissolved and microbial pool, how do they compare to usual concentration of DOM and microbial biomass? I assume they were equal for all moisture and temperature treatments, right?

P3l20: Citation of the kinetic respiration analysis (Wutzler 2011) is not appropriate in this context. I assume you wanted to refer to: Wutzler T & Reichstein M (2008) Colimitation of decomposition by substrate and decomposers - a comparison of model formulations. Biogeosciences, 5, 749-759 10.5194/bg-5-749-2008

Fig 4: There seem to be two groups of observations, a higher branch and a lower one. Why is this? Is it ok to fit a single smoother to this data?

Technical comments:

The grammar of the paper needs to be re-checked, e.g. p9L18, p9L25, p13L19.

---

## Author Comment (AC2) · 16 May 2018

We thank T. Wutzler for a constructive review. Below we address each comment individually.

Reviewer comments are quoted followed by the author response and, where relevant, changes to the manuscript.

————————

"Moyano et al. compare several versions of SOM turnover models with a comprehensive set of observations of varying temperature and soil moisture. They show that explicitly accounting for diffusion, compared to using empirical formulation of the temperature/moisture rate modifiers, improves fit and understanding of SOM decomposition. This result is interesting to the soil model developers and biogeoscientists studying SOM turnover and consequences at soil core to larger scales. The paper contains a strong validation by a good agreement with independent data. The clarity of the discussion on reasons for the good validation fit, interactions with initial pools, and different resulting temperature sensitivities can be improved. With an extension of discussion and some more clarifications in the discussion, the paper could be published. Nevertheless, I suggest several additional tasks with this model and data, that would help the community."

Please see responses to comments by R. Grant for changes already made to the discussion.
* * *
"First, while the paper already contains three different structural versions of decomposition, I suggest including another version of an inverse Michaelis-Menten dynamics for depolymerization (but not for DOM uptake), where the non-linear term is in enzymes instead of the substrate (F = V C_P * C_E/(K + C_E)). This would broaden the application of conclusions of this study, because the inverse formulation is used by many microbial models since suggested by Schimel and Weintraub 2003."

Reverse MM kinetics assume that enzyme concentrations can increase enough that they start to compete for binding sites on SOM and thus saturate at some point. Schimel and Weintraub used this approach to deal with a problem of model instability driven by the dynamics of the microbial pool. However, we think a general saturation of the available SOM by enzymes in soils is unlikely to be the norm, as it would imply a large and likely unsustainable production of enzymes and very rapid decomposition of all polymeric C. We did not have stability issues in our model that would justify using this MM form and find it an unlikely explanation of soil C dynamics. However, we

will test the effect of using the reverse MM and, if relevant, include information in the revised manuscript.

Changes in manuscript: inclusion of any relevant results following model calibration using reverse-MM and comparison with other versions.

———————————

"Second, the study describes a decoupling between depolymerization and microbial uptake at low diffusion rates, I assume by accumulating OM in the dissolved pool. Are the fluxes correlated again for the same treatment, if you aggregate over say two weeks? The decoupling is a challenging fact for upscaling studies, that often assume the DOM pool in quick quasi steady state with decomposition and microbial uptake. For low moisture the decomposed flux was almost not taken up and respired (Fig. 9). Would this also be true with two separate DOM pools after longer time? I would appreciate an extended discussion on this topic."

As well pointed out, Figure 9 shows that for some samples at lower moisture decomposition did not equilibrate with uptake in the 6 months of the simulated incubation. The plot also gives a good idea of how fast soil at higher moisture content return to equilibrium. We kept the model simple where possible. The current form where there is only one DOC pool is a simplification that assumes microbes have access to an amount equivalent to the concentration in the bulk soil times a conductivity value. If conductivity is not 0, this amount will increase if the concentration increases, until the input from decomposition equals output from uptake. The reason this is done differently for enzymes is that enzymes have a decay rate, which means that the pool decreases with time. So even if equilibrium is reached, the flux of enzymes from microbes to the decomposition site will be lower if conductivity is lower, simply because a larger fraction is lost before diffusing. Further analysis of the model could indeed go more into detail looking at such dynamics. We take this as a suggestion future research.

Changes in manuscript: discussion extended under section 7.2 Moisture effects and

diffusion limitation

————————

"The model used enzyme pools split to locations but a simplified diffusion limited rate multiplier for DOM. What is the reasoning for this decision, and what are the expected consequences for using a rate modifier for enzymes too?"

See response to previous comment.

————————

"There is an interesting differentiation between parameterized temperature sensitivity (E_a) and an apparent predicted one, the latter one also depending on partitioning of the pools (P13l20ff). What are the reasons and consequences here. The paper would profit from an extended discussion here."

We have extended the discussion on this topic. See also responses to R. Grant.

Changes in manuscript: discussion extended. P.14 L.2-18

————————

"p5l10: The choice of the wording "particulate" suggests to OM floating together with the DOM. I assume instead that C_P comprises litter and residues also sitting on surfaces. When using "polymeric" it conveys a different connotation and still the "P" can be used as acronyms."

We followed the advice and changed to "polymeric". (For the record: according to Wikipedia "Particulate organic matter is defined as soil organic matter between 0.053 mm and 2 mm in size".)

Changes in manuscript: the term "particulate" was changed to "polymeric"

————————

"p7l10: The model assumes enzyme production to be modeled similar to growth respiration as a fraction of uptake, instead similar to maintenance respiration as determined by microbial biomass. What are the reasons for this formulation ? "

From a practical side, initial testing of model structure resulted in this approach fitting the data best (data not shown). From a theoretical side, is would be logical that microbes produce enzymes mostly when C becomes available and save resources otherwise. A continued enzyme production would lead to an unnecessary depletion of resources. We now note this in the same paragraph.

Changes in manuscript: text added near P.7 L.10-12

————————

"p8l4ff: The wording here suggests, that all the processes have the same temperature sensitivity, i.e. same $E_a$. I suggest adding another index to $E_a$ that this parameter varies between processes."

We followed the suggestion and added a subindex.

Changes in manuscript: $E_a$ in equation 19 changed to $E_{ap}$

————————

"P8l10: I assume there is only one set of parameters fitted to the entire data of all temperature and moisture treatments. Would be nice to state that here. Please, also state the number of fitted parameters, and add the initial partitions to Table 1."

That is correct. This is now clarified. Because of space limitations, initial parameter values and their lower and upper bounds were added in a table in the supplementary material.

Changes in manuscript: - text added at P.8 L.10-11. - table with initial parameter values and boundaries added to supplementary material.

————————

"The fitted parameter vector in a 20 dimension space is quite challenging for a gradient based search. Did you check global convergence by starting from more states, maybe more random distributed as just the one described for p11l30. What are the most important correlations in parameter estimates?"

We agree. A global minima can of course not be guaranteed. In preliminary work we explored parameter spaces manually and using latin hyper square methods. The initial parameter values used here are already the result of these tests. Parameters in such model are often correlated and this was also the case in our study. High, but not very high, correlations occurred between some parameters, e.g. V_U_ref and g_0 (0.89), V_U_ref and E_r (0.83), V_D_ref and g_0 (0.84), f_CD and E_K (0.83) and V_D_ref and V_U_ref (0.8), f_ug and f_ge (0.83). We therefore do not make conclusions on how well constrained our estimates are, as this information is not obtained with gradient or deterministic algorithms. However, we now added these remarks and a correlation plot in the supplementary material for extra information.

Changes in manuscript: - text added in the discussion P.12 L. 13-16 - correlation plot of parameter sensitivities added to the supplementary material

―――――――――――

"P8l10: model calibration open questions: How were enzyme pools initialized? What were the values of fractions for particulate, dissolved and microbial pool, how do they compare to usual concentration of DOM and microbial biomass? I assume they were equal for all moisture and temperature treatments, right?"

As clarified in response to comments by R. Grant, all C pools were initialized by fitting them similarly to other model parameters (as initial steady state was not assumed). Upper and lower bounds were set (see Table S1) to assure they stayed in a realistic range (text added in P.8 L.23-24). The initial values of these fractions are found in Table 1. fM with 0.07 is in particular on the upper range of observed values.

—————————

"P3l20: Citation of the kinetic respiration analysis (Wutzler 2011) is not appropriate in this context. I assume you wanted to refer to: Wutzler T & Reichstein M (2008) Colimitation of decomposition by substrate and decomposers - a comparison of model formulations. Biogeosciences, 5, 749-759 10.5194/bg-5-749-2008"

Thanks for pointing this out.

Changes in manuscript: reference changed to Wutzler T & Reichstein M (2008)

—————————

"Fig 4: There seem to be two groups of observations, a higher branch and a lower one. Why is this? Is it ok to fit a single smoother to this data?"

Figure 4 is mainly meant as a visual aid since it is not possible to mark which model point corresponds to which data point. The smooth lines ignore the variability along the y axis, caused mainly by the time effects resulting from two incubation cycles, but they help visualize the general resulting relationship between moisture content and respiration fluxes. We added this clarification in the results section.

Changes in manuscript: added text near P.11 L.2-4

—————————

"Technical comments: The grammar of the paper needs to be re-checked, e.g. p9L18, p9L25, p13L19."

Changes in manuscript: spelling and grammar mistakes were corrected.

—————————

Please also note the supplement to this comment:
https://www.biogeosciences-discuss.net/bg-2018-95/bg-2018-95-AC2-supplement.pdf

[Figure]

**Supplement:**

**Model M2-dif steady state equations**

The equilibrium solutions to the C pools of model M2-dif are given by:

$$C_P = K_D r_{ed} z(-2gI_{ml}f_{ge}f_{ug}r_{md} + 2gI_m f_{ug}r_{md} - 2gI_{sl}f_{ge}r_{mr}f_{ug} - 2gI_{sl}f_{ge}f_{ug}r_{md} + 2gI_{sl}r_{mr} +$$
$$2gI_{sl}r_{md} - I_{ml}f_{ge}f_{ug}r_{ed}r_{md} + I_{ml}f_{ug}r_{ed}r_{md} - I_{sl}f_{ge}r_{mr}f_{ug}r_{ed} - I_{sl}f_{ge}f_{ug}r_{ed}r_{md} + I_{sl}r_{mr}r_{ed} +$$
$$I_{sl}r_{ed}r_{md})/(gI_{ml}V_D f_{ge}r_{mr}f_{ug} + gI_{ml}V_D f_{ge}f_{ug}r_{md} + 2gI_{ml}f_{ge}f_{ug}r_{ed}r_{md} - 2gI_{ml}f_{ug}r_{ed}r_{md} +$$
$$gI_{sl}V_D f_{ge}r_{mr}f_{ug} + gI_{sl}V_D f_{ge}f_{ug}r_{md} + 2gI_{sl}f_{ge}r_{mr}f_{ug}r_{ed} + 2gI_{sl}f_{ge}f_{ug}r_{ed}r_{md} - 2gI_{sl}r_{mr}r_{ed} -$$
$$2gI_{sl}r_{ed}r_{md} + I_{ml}f_{ge}f_{ug}r_{ed}^2 r_{md} - I_{ml}f_{ug}r_{ed}^2 r_{md} + I_{sl}f_{ge}r_{mr}f_{ug}r_{ed}^2 + I_{sl}f_{ge}f_{ug}r_{ed}^2 r_{md} - I_{sl}r_{mr}r_{ed}^2 -$$
$$I_{sl}r_{ed}^2 r_{md}) \tag{A1}$$

$$C_D = -z(r_{mr} + r_{md})/(gV_U f_{ug}(f_{ge} - 1)) \tag{A2}$$

$$C_M = f_{ug}(I_{ml}f_{ge} - I_{ml} + I_{sl}f_{ge} - I_{sl})/(f_{ge}r_{mr}f_{ug} - r_{mr} + f_{ug}r_{md} - r_{md}) \tag{A3}$$

$$C_{ED} = -gf_{ge}f_{ug}(I_{ml}r_{mr} + I_{ml}r_{md} + I_{sl}r_{mr} + I_{sl}r_{md})/(r_{ed}(2gf_{ge}r_{mr}f_{ug} - 2gr_{mr} + 2gf_{ug}r_{md} -$$
$$2gr_{md} + f_{ge}r_{mr}f_{ug}r_{ed} - r_{mr}r_{ed} + f_{ug}r_{ed}r_{md} - r_{ed}r_{md})) \tag{A4}$$

$$C_{EM} = -f_{ge}f_{ug}(gI_{ml}r_{mr} + gI_{ml}r_{md} + gI_{sl}r_{mr} + gI_{sl}r_{md} + I_{ml}r_{mr}r_{ed} + I_{ml}r_{ed}r_{md} + I_{sl}r_{mr}r_{ed} +$$
$$I_{sl}r_{ed}r_{md})/(r_{ed}(2gf_{ge}r_{mr}f_{ug} - 2gr_{mr} + 2gf_{ug}r_{md} - 2gr_{md} + f_{ge}r_{mr}f_{ug}r_{ed} - r_{mr}r_{ed} + f_{ug}r_{ed}r_{md} -$$
$$r_{ed}r_{md})) \tag{A5}$$

In these equations, $I_{ml}$ and $I_{sl}$ are metabolic and structural litter input, which represent litter additions to the $C_D$ and $C_P$ pools, respectively.

**Supplementary figures**

[Figure]

**Figure S1: Relationships of apparent activation energies against soil volumetric moisture content. Values are given for measured and modelled respiration and for modelled decomposition. Each plot compares observed values against a different calibrated model (M1-dif, M2-dif and MM-dif). Apparent activation energies are shown for the temperature ranges 5-20 (top panel) and 20-35 °C (bottom panel).**

[Figure]

**Figure S2: The relationship between respiration rates and soil moisture content shown for measured and modelled values. Each plot compares the measurements a different model (M1-dif, M2-dif, MM-dif). Lines are a smooth loess fit to show the average relationship.**

[Figure]

**Figure S3: Respiration (growth and maintenance) and decomposition fluxes modelled using M2-dif against soil moisture (left plot) and soil temperature (right plot). Shaded areas denote the 95% confidence intervals from a loess fit.**

[Figure]

**Figure S4: Correlations between sensitivity functions of model parameters (obtained by using R function sensFun from package FME). All R² values are below 0.9 but several are above 0.8, e.g. between V_U_ref and g_0 (0.89), V_U_ref and E_r (0.83), V_D_ref and g_0 (0.84), f_CD and E_K (0.83) and V_D_ref and V_U_ref (0.8), f_ug and f_ge (0.83).**

10

Table S1: Calibrated model parameters with initial values and lower and upper bounds.

| Name | Units | Initial value | Lower bound | Upper bound | Calibrated value |
|------|-------|---------------|-------------|-------------|------------------|
| $g_0$ | $h^{-1}$ | 1.4 | 0.1 | 10 | 2.2 |
| $E_{a\_K}$ | kJ | 90 | 10 | 130 | 89 |
| $E_{a\_r}$ | kJ | 90 | 10 | 130 | 95 |
| $E_{a\_V}$ | kJ | 90 | 10 | 130 | 87 |
| $f_D$ | - | 8.8E-05 | 1E-05 | 0.001 | 9.2e-5 |
| $f_E$ | - | 0.00058 | 1E-05 | 0.001 | 6.1e-4 |
| $f_M$ | - | 0.043 | 0.001 | 0.1 | 0.071 |
| $f_{ug}$ | - | 0.6 | 0.3 | 0.8 | 0.7 |
| $f_{ge}$ | - | 0.025 | 0.01 | 0.1 | 0.025 |
| $K_{D\_ref}$ | kg C m$^{-3}$ | 60 | 30 | 300 | 50 |
| $n$ | - | 2.7 | 1 | 3 | 2.3 |
| $m$ | - | 1.2 | 1 | 3 | 1.2 |
| $r_{ed\_ref}$ | $h^{-1}$ | 0.0002 | 1E-05 | 0.001 | 5e-4 |
| $r_{md\_ref}$ | $h^{-1}$ | 0.002 | 0.001 | 0.01 | 1.5e-3 |
| $r_{mr\_ref}$ | $h^{-1}$ | 4E-05 | 1E-06 | 0.001 | 4.2e-5 |
| $V_{D\_ref}$ | $h^{-1}$ | 0.3 | 0.1 | 1 | 0.35 |
| $V_{U\_ref}$ | $h^{-1}$ | 0.09 | 0.01 | 0.1 | 0.092 |

---

## Author Response (AR1)

Dear Editor,

We have carried out a thorough revision of our manuscript. Most importantly, this included a recalibration of all models following a well documented procedure. The methods have been updated to reflect this and provide all steps to reproduce the study. In the new calibration we carried out a well documented first step to explore parameter spaces using Latin hypersquare sampling. In addition, we changed the following:

- Activation energy parameters for all reaction rates were merged into one parameter E_V (not for enzyme and microbial decay).
- Parameter f_ug was fixed at 0.7 following Hagerty et al. 2014.
- V_D was separated into different parameters, one for each reaction kinetics type (Eq. 6-9). The latter was done after calculating that the ranges required for these parameters differ by several orders of magnitude, and thus require separate calibration (our previous calculations had been incorrect).

Recalibration generally resulted in a better model fit (an exception was model M2-sat which we took anyways for consistency). In particular, first and second order decomposition models now resulted in good fits, so that we were able to make a relevant comparison between them and Michaelis-Menten models. The manuscript was thus generally and significantly improved. The main results remained unchanged, with diffusion and MM kinetics performing the best. All figures and numbers were updated. A second temperature sensitivity figure was added. Several figures and tables were also added to the supplementary material.

The improved results allowed us to better address the comments from the reviewers (as discussed in our respective answers to each). Thus, we added a new model version with reverse Michaelis-Menten kinetics and expanded the results and discussion sections on temperature responses as well as several other sections.

We also changed the ms title to reflect the broader focus of the study: "Diffusion limitations and Michaelis-Menten kinetics as drivers of combined temperature and moisture effects on carbon fluxes of mineral soils"

Here follow specific responses to editor comments and our previous responses to reviewers added below. (Note that pages and lines of modifications are respect to the non-revised manuscript. Positions may differ in the new version).

*1. When measuring the result of the cumulated CO2 evolution in differing time intervals, i.e. in sealed samples, the O2 concentration will fall under the experiment, accordingly.*
*Was the reduction in O2 concentrations considered and if not, what were the consequences for the interpretation of the results?*

According to the accumulated CO2, minimum O2 levels were over 15 %. We now discuss O2 limitations in the manuscript in light of previous literature and our observations. P16 L9-14 (revised ms)

*2. In your comment to Robert Grant's report, you mention (page 3 at the end of the second paragraph):*
*"Adding further complexity would introduce parameter identifiability problems and add no relevant information."*
*If you intend to use this argument in the discussion, I would ask you, to consider the situation that two or more drivers affect the same process in a similar way. Examining only one of them, will*

*include the risk of falling for a "wrong" one in the empirical analysis. So, one should be careful when going for simplicity and discuss the risks of doing so.*

We agree. With "no relevant information" we meant functions with variables that in our experiment do not change. While O2 is a changing factor, we now discuss why it may not have been limiting. See response to comments below.

*3. In your comments to Thomas Wutzler's report, you mention on page 6 that you based your parameter space exploration and choice of initial parameters on preliminary work ('manual'...) . I would rather recommend using the appropriate statistical methods (as suggested in the report) and document the robustness of the approach. Then give the corresponding parameter correlation matrix (appendix) and discuss the main parameter correlations in the text also in the light of model complexity and equifinality (see point mentioned above). Please make sure to use a methodology that is reproducible (the 'manual' is not).*

As stated above, models were all recalibrated following a documented procedure. Using a Latin hypercube of parameter spaces as a first step allowed a better exploration of parameter spaces. See manuscript for details. We now added the parameter correlation matrix as well as kernel density estimation plots to indicate which parameters were better constrained.
* * *
Response to comments by R. Grant

We thank R. Grant for a critical evaluation of our work. Below we address each comment individually.
* * *
*"General comments This is an interesting model study that makes a key point that higher order kinetics are needed to model respiration responses to changes in temperature and soil water content. This point is important because many SOC models still retain first order kinetics when making projections of climate change effects on SOC and hence on climate feedbacks, with a possible risk of error. Of particular interest to me but not often considered in modelling was the reduced sensitivity to temperature of microbial decay vs that of uptake and growth, and the point raised in the Discussion about density-dependent microbial decay, as both these processes affect microbial biomass and hence decomposition rates in higher order models."*

We agree. We found it particularly interesting that from a process perspective we can demonstrate that respiration dynamics do not necessarily reflect decomposition dynamics in the short and mid-term.
* * *
*"The authors have done a good job in testing model performance against experimental results from soil incubations. However a more constrained testing of its performance would be achieved by comparing the actual time courses of CO2 emissions measured during the incubations, rather than just the totals measured over the duration of each treatment as done in this study."*

Soil respiration was determined by measuring the accumulated $CO_2$ in the flask's headspace at irregular intervals. As such, the data represent a time course, although irregular and of low frequency (we unfortunately did not have the capability to measure high frequency respiration rates). This actual measured data was what we used for modelling purposes. While this is already

described in the methods, we added text in order to make the procedure clearer.

Changes in manuscript: added text near P.4 L.28-29, P.8 L.10-14
* * *
*"Two key areas need to be developed for this model to be capable of more robust performance and hence wider application: the coupling of C with N for all transformations, because kinetics of decomposition and respiration are strongly affected by SOC quality, and the simulation of O2 limitations on microbial activity rather than fitting declines in activity with higher soil water contents. Both these areas are already well developed in some ecosystem models."*

We fully agree that these two aspects are potentially important for building predictive soil carbon models for general application. There are other processes that should be included in predictive models as well, such as oregano-mineral associations and vertical transport of SOM/DOC. However, this study has a defined focus and purposefully ignores many process that could be relevant under a variety of situations. This study does not present a generally valid predictive model. On the other hand, we believe the model showed a robust performance for the purpose of simulating our observations. Diffusion limitations as implemented here can, in future studies, be integrated in more complex predictive models and validated against larger sets of data.

The variability in SR at high water content was captured by our diffusion-based model without a representation of O2 limitation. We note that this model does not include an empirical decline at higher moisture (as we understand is suggested in the comment). Only the saturation function in the alternative model we compare it with included such an empirical function. This does not imply that O2 limitations are not an important limiting factor in saturated soils and mechanistic simulations in that area are useful.

In summary, our approach was to start from a simple model and add complexity until the observations could be reproduced, specifically testing the effect of adding diffusion. Adding further complexity would introduce parameter identifiability problems and add no relevant information. That said, we understand the comment addresses the *"wider application"* of the model and we now further address this by extending the discussion.

Changes in manuscript: added text near P.13 L.1-7
* * *
*"P.2. L.15: a strong effects ? Introduction P.2 L.29: But note more rapid soil N mineralization, uptake, NPP and litterfall that may offset this feedback. N cycling very much needs to be included in any soil SOC model and the authors need to acknowledge this."*

We now acknowledge this in the discussion making the model limitations more explicit.

Changes in manuscript: added text near P.13 L.15-18
* * *
*"P.4 L.16: Soil grinding and mixing will increase microbial access to SOC from that in a natural soil, likely raising decomposition rates."*

Agreed but unavoidable in this setup. To minimize such effects, we allowed samples to rest during pre-incubation.
* * *
*"P.4 L.18: A BD of 1.8 is inconsistent with a porosity of 0.45. One or the other must be checked."*

Bulk density was incorrect and was changed.

Changes in manuscript: bulk density corrected to 1.4 g cm-3. P.4 L.18

\----------------------------
*"P.4 L.21: The saturated value of 0.25 is less than the porosity of 0.45."*

Please note these are gravimetric moisture values, so 0.25 in g/g is close to the 0.45 volumetric content.

\----------------------------
*"P.5 L.6: This statement is valid as long as the model simulates experimental protocol (e.g. duration of treatments)."*

Agreed. Our simulations reproduced the exact incubation protocols we used.

\----------------------------
*Modelling approach*
*"P.7 L.3: Although if diffusion limitations reduce FPD you will also reduce CD and hence uptake, so make sure there isn't a duplicated effect caused by direct diffusion limitation to CD."*

Since in the model diffusion affects enzyme pools and the availability of CD for uptake, there is a double effect on uptake, one indirect and one direct. This is intended. However, because CE decays and CD does not, long term effects will result from the former but not from the latter limitation. Unless CD is lost through another path such as leaching (here not considered).

\----------------------------
*"P..7 L.7: Check variable names."*

Changes in manuscript: corrected the variable name.

\----------------------------
*"P.7 L.13: But temperature sensitivity of fmr is different from that of growth."*

Equations 15 and 16 had not been updated to the latest model version and were incorrect. They have been corrected. (But note that other manuscript sections were correct, e.g. parameter table). r_mr is in fact temperature dependent, as expected. Temperature dependencies were calibrated, since we did not find strong evidence for fixing these parameters.

Changes in manuscript: corrected a wrong version of equations 15 and 16. Now F_MP= C_M * r_md and F_MRM = C_M * r_mr

\----------------------------
*"P.8. eq.19: Low and high temperature inactivation terms are often used with Arrhenuis equations to give greater Q10 at low temperature and must lower Q10 at higher while using biologically realistic values of Ea (typically ca 65 kJ mol-1)."*

With a simple Arrhenius function, we found that the observed Q10 can vary between low and high temperatures (see supplementary figures). For the range we used, between 4 and 35 degC, a more complex temperature function was not justified.

\----------------------------

*Model Calibration*
*"Was a spinup run used to enable key state variables to stabilize at values independent of those initialized? This is standard modelling protocol."*

A spinup would be valid only if a steady state at initial conditions is assumed. Our soil was from arable fields and pre-processes in the lab. Because of this we did not assume steady state at time 0 and instead estimated the initial pool sizes through calibration, as done in other studies (Menichetti, et al., *Biogeosciences,* 2016). This is stated in P.8 L.14-16.
* * *
*"P.8 L.25: This is a commendable objective because some SOC models still retain first order algorithms."*

Agreed.
* * *
*"P.9 eq. 20, 21: Reductions of f (theta) and f(psi) at higher theta and psi are caused by O2 deficiency as noted later in the text, and are better modelled as such because these reductions are temperature-dependent."*

We commented on the O2 limitations above and in the paper discussion.
* * *
*"P.10 L.4: MPa Results"*

Changes in manuscript: spelling corrected
* * *
*"P.11 L.6: How were C inputs evaluated, as in natural ecosystems these also vary with temperature and swc. In fact, these inputs are the most important part of a SOC model as they are the main drivers of microbial activity."*

We used a fixed value of 1.2 g d$^{-1}$ C, which we found to be realistic for cultivated temperate soils. Steady state was calculated analytically (supplementary equations) so inputs, temperature and swc needed to be constants, as described.
* * *
*"P.11 L.13-14. This is a nice test of the model. Describe how values for initialization of C pools and threshold swc were determined for this study. How did these values affect model results, particularly without model spinup? Ideally you should just change total SOC as determined from the soil measurement, and develop rules for allocating total SOC to initial C pools depending on site conditions, and then spinning up the model to equilibrium before comparison with observed values. An even better test of the model would be against the actual time course of CO2 effluxes measured during each incubation, as has been done in earlier modelling studies (e.g. Soil Sci. Soc. Amer. J. 58:1681-1690). This test lets you see whether the model is really simulating the temporal dynamics of respiration at different water contents under changing temperatures."*

As stated in P.9 L.23 – P.10 L.4., the C pools were initialized not by spinup but by calibration, given that also in the validation case we could not determine if initial conditions were in steady state. The reason to calibrate the swc threshold during validation is that this parameter is expected to change between soils but we do not currently have a reliable means to estimate it, as stated in P.10 L.1-4. We expanded the discussion where we address the issue of how pool sizes may be

affecting the modelled and observed values

Changes in manuscript: text added near P.14 L.2–8
* * *
Discussion
*"P.12 L.10. Specify these changes as noted in Results to establish how robust the model really is."*

P.12 L.10 reads "We note that few studies were found with data on moisture and temperature interactions and this was the only validation attempt carried out." It is not clear what changes are referred to.
* * *
*"P.12 L.19-20. Would this problem be addressed by a cold temperature inactivation term in eq. 19?"*

A further decrease in activity using an inactivation term, while realistic, would probably exacerbate the problem here, since it seemingly already is the result of the lower rates under colder conditions. A solution to this problem is however out of the scope of this study.
* * *
*"P.12 L.29-30. The absence of O2 limitations is likely causing the reductions in Ea and Q10 at higher swc in Fig. 5. Modelling these limitations should be a key next step in model development. These limitations are already simulated in some other ecosystem models."*

We believe it is likely not a O2 limitation for two reasons. First, the decrease occurs sharply at ca. 50% saturation. At this water content and in small samples O2 should not be limiting. Second, our model reproduced this decrease quite well without O2 limitations, showing that it is the result of pool dynamics.
* * *
*"P.13 L.4-5: The reduced temperature sensitivity of microbial and enzyme decay needed to model realistic biomass at different temperatures is an important finding of this study."*

Yes, our results are compatible with such lower values.
* * *
*"P.13 L.15: Experimental determinations of Ea are often in the 65 kJ mol-1 range. The larger value modelled here may have been required in the absence of a cold tempera- ture inactivation term in eq. 19."*

If 4 degC is "cold" then this may be the case. It should be noted, however, that experimental determinations are "apparent" values. Apparent values given by our model are also in the lower range.
However, our focus is more on the distinction between the prescribed values (parameter values) and the apparent ones and how these may interact with moisture. A detailed analysis of temperature effects is outside this study's scope.

Changes in manuscript: added text near P.14 L.2-8
* * *
*"P.13 L.16-18. Lower values probably arise from O2 limitations. The authors realistically address the current limitations of the model."*

We do not discard a O2 effect, especially near saturation. But see our responses above.
* * *
*"P.13 L.32: models"*

Changes in manuscript: spelling corrected.
* * *
*"P.14 L.8-9. Why not make the percolation threshold depend on soil water potential (e.g. -15 MPa)? This might improve model robustness by reducing reparameterization for each soil."*

The reason is that the published value was not valid for this soil, as explained in the following lines P.14 L.22-25.
* * *
*"P.14 L.30-21: Would CUE decline at higher temperatures if Rm (fmr in (16)) increased exponentially with temperature, as it is known to do?"*

As noted above, the equations using fmr were outdated and have been corrected. The parameter r_mr is temperature dependent and determines the Rm flux. If CUE changes with T may depend on its definition. Here we define it as f_ug, so it remains constant.
* * *
We thank T. Wutzler for a constructive review. Below we address each comment individually.

Reviewer comments are in quotations followed by the author response and, where relevant, changes to the manuscript.

Response to comments by T. Wutzler

*"Moyano et al. compare several versions of SOM turnover models with a comprehensive set of observations of varying temperature and soil moisture. They show that explicitly accounting for diffusion, compared to using empirical formulation of the temperature/moisture rate modifiers, improves fit and understanding of SOM decomposition. This result is interesting to the soil model developers and biogeoscientists studying SOM turnover and consequences at soil core to larger scales. The paper contains a strong validation by a good agreement with independent data. The clarity of the discussion on reasons for the good validation fit, interactions with initial pools, and different resulting temperature sensitivities can be improved. With an extension of discussion and some more clarifications in the discussion, the paper could be published. Nevertheless, I suggest several additional tasks with this model and data, that would help the community."*

Please see responses to comments by R. Grant for changes already made to the discussion.
* * *
*"First, while the paper already contains three different structural versions of decomposition, I suggest including another version of an inverse Michaelis-Menten dynamics for depolymerization (but not for DOM uptake), where the non-linear term is in enzymes instead of the substrate (F = V*

*C_P \* C_E/(K + C_E)). This would broaden the application of conclusions of this study, because the inverse formulation is used by many microbial models since suggested by Schimel and Weintraub 2003.”*

Reverse MM kinetics assume that enzyme concentrations can increase enough that they start to compete for binding sites on SOM and thus saturate at some point. Schimel and Weintraub used this approach to deal with a problem of model instability driven by the dynamics of the microbial pool. However, we think a general saturation of the available SOM by enzymes in soils is unlikely to be the norm, as it would imply a large and likely unsustainable production of enzymes and very rapid decomposition of all polymeric C. We did not have stability issues in our model that would justify using this MM form and find it an unlikely explanation of soil C dynamics. However, we will test the effect of using the reverse MM and, if relevant, include information in the revised manuscript.

Changes in manuscript: inclusion of any relevant results following model calibration using reverse-MM and comparison with other versions.
* * *
*“Second, the study describes a decoupling between depolymerization and microbial uptake at low diffusion rates, I assume by accumulating OM in the dissolved pool. Are the fluxes correlated again for the same treatment, if you aggregate over say two weeks? The decoupling is a challenging fact for upscaling studies, that often assume the DOM pool in quick quasi steady state with decomposition and microbial uptake. For low moisture the decomposed flux was almost not taken up and respired (Fig. 9). Would this also be true with two separate DOM pools after longer time? I would appreciate an extended discussion on this topic.”*

As well pointed out, Figure 9 shows that for some samples at lower moisture decomposition did not equilibrate with uptake in the 6 months of the simulated incubation. The plot also gives a good idea of how fast soil at higher moisture content return to equilibrium.
We kept the model simple where possible. The current form where there is only one DOC pool is a simplification that assumes microbes have access to an amount equivalent to the concentration in the bulk soil times a conductivity value. If conductivity is not 0, this amount will increase if the concentration increases, until the input from decomposition equals output from uptake. The reason this is done differently for enzymes is that enzymes have a decay rate, which means that the pool decreases with time. So even if equilibrium is reached, the flux of enzymes from microbes to the decomposition site will be lower if conductivity is lower, simply because a larger fraction is lost before diffusing. Further analysis of the model could indeed go more into detail looking at such dynamics. We take this as a suggestion future research.

Changes in manuscript: discussion extended under section 7.2 Moisture effects and diffusion limitation
* * *
*“The model used enzyme pools split to locations but a simplified diffusion limited rate multiplier for DOM. What is the reasoning for this decision, and what are the expected consequences for using a rate modifier for enzymes too?”*

See response to previous comment and addition to discussion.
* * *
*“There is an interesting differentiation between parameterized temperature sensitivity (E_a) and an apparent predicted one, the latter one also depending on partitioning of the pools (P13l20ff). What are the reasons and consequences here. The paper would profit from an extended discussion here.”*

We have extended the discussion on this topic. See also responses to R. Grant.

Changes in manuscript: discussion extended. P.14 L.2-18
* * *
*"p5l10: The choice of the wording "particulate" suggests to OM floating together with the DOM. I assume instead that C_P comprises litter and residues also sitting on surfaces. When using "polymeric" it conveys a different connotation and still the "P" can be used as acronyms."*

We followed the advice and changed to "polymeric". (For the record: according to Wikipedia "Particulate organic matter is defined as soil organic matter between 0.053 mm and 2 mm in size".)

Changes in manuscript: the term "particulate" was changed to "polymeric"
* * *
*"p7l10: The model assumes enzyme production to be modeled similar to growth respiration as a fraction of uptake, instead similar to maintenance respiration as determined by microbial biomass. What are the reasons for this formulation ? "*

From a practical side, initial testing of model structure resulted in this approach fitting the data best (data not shown). From a theoretical side, is would be logical that microbes produce enzymes mostly when C becomes available and save resources otherwise. A continued enzyme production would lead to an unnecessary depletion of resources. We now note this in the same paragraph.

Changes in manuscript: text added near P.7 L.10-12
* * *
*"p8l4ff: The wording here suggests, that all the processes have the same temperature sensitivity, i.e. same E_a. I suggest adding another index to E_a that this parameter varies between processes."*

We followed the suggestion and added a subindex.

Changes in manuscript: $E\_a$ in equation 19 changed to $E\_{ap}$
* * *
*"P8l10: I assume there is only one set of parameters fitted to the entire data of all temperature and moisture treatments. Would be nice to state that here. Please, also state the number of fitted parameters, and add the initial partitions to Table 1.*

That is correct. This is now clarified. Because of space limitations, initial parameter values and their lower and upper bounds were added in a table in the supplementary material.

Changes in manuscript:
   - text added at P.8 L.10-11.
   - table with initial parameter values and boundaries added to supplementary material.
* * *
*The fitted parameter vector in a 20 dimension space is quite challenging for a gradient based search. Did you check global convergence by starting from more states, maybe more random distributed as just the one described for p11l30. What are the most important correlations in*

*parameter estimates?"*

We agree. A global minima can of course not be guaranteed. In preliminary work we explored parameter spaces manually and using latin hyper square methods. The initial parameter values used here are already the result of these tests. Parameters in such model are often correlated and this was also the case in our study. High, but not very high, correlations occurred between some parameters, e.g. V_U_ref and g_0 (0.89), V_U_ref and E_r (0.83), V_D_ref and g_0 (0.84), f_CD and E_K (0.83) and V_D_ref and V_U_ref (0.8), f_ug and f_ge (0.83). We therefore do not make conclusions on how well constrained our estimates are, as this information is not obtained with gradient or deterministic algorithms. However, we now added these remarks and a correlation plot in the supplementary material for extra information.

Changes in manuscript:
  - text added in the discussion P.12 L. 13-16
  - correlation plot of parameter sensitivities added to the supplementary material
* * *
*"P8l10: model calibration open questions: How were enzyme pools initialized? What were the values of fractions for particulate, dissolved and microbial pool, how do they compare to usual concentration of DOM and microbial biomass? I assume they were equal for all moisture and temperature treatments, right?"*

As clarified in response to comments by R. Grant, all C pools were initialed by fitting them similarly to other model parameters (as initial steady state was not assumed). Upper and lower bounds were set (see Table S1) to assure they stayed in a realistic range (text added in P.8 L.23-24). The initial values of these fractions are found in Table 1. $f_M$ with 0.07 is in particular on the upper range of observed values.
* * *
*"P3l20: Citation of the kinetic respiration analysis (Wutzler 2011) is not appropriate in this context. I assume you wanted to refer to: Wutzler T & Reichstein M (2008) Colimitation of decomposition by substrate and decomposers - a comparison of model formulations. Biogeosciences, 5, 749-759 10.5194/bg-5-749-2008"*

Thanks for pointing this out.

Changes in manuscript: reference changed to Wutzler T & Reichstein M (2008)
* * *
*"Fig 4: There seem to be two groups of observations, a higher branch and a lower one. Why is this? Is it ok to fit a single smoother to this data?"*

Figure 4 is mainly meant as a visual aid since it is not possible to mark which model point corresponds to which data point. The smooth lines ignore the variability along the y axis, caused mainly by the time effects resulting from two incubation cycles, but they help visualize the general resulting relationship between moisture content and respiration fluxes. We added this clarification in the results section.

Changes in manuscript: added text near P.11 L.2-4
* * *
*Technical comments:*

*The grammar of the paper needs to be re-checked, e.g. p9L18, p9L25, p13L19."*

Changes in manuscript: spelling and grammar mistakes were corrected.

[revised manuscript text omitted]

---

## Author Response (AR2)

Response to second review by R. Grant

Below we address each point.

5    *p. 4 l. 17: the maximum water content of 25% w/w would correspond to one of 35%v/v with a BD of 1.4, less that the porosity of 45% v/v, so that full saturation at which O2 limitations would be expected may not have been reached.*
Indeed, here the numbers did not match. These w/w values were not updated after correcting the bulk density in our first revision (1.8 to 1.4 g/cm3) and thus were still wrong.
10   To simplify and avoid confusion, we now simply show volumetric values.

*p. 8 l. 15: Initialization of C pool sizes without spinup thereafter makes model output an artefact of this initialization, so that re-initialization with a different soil makes detracts from the independence needed in validation.*
15   We understand the concern but do not fully agree. Optimizing initial pool sizes indeed gives the model more freedom to fit the data. But the outcome is not an 'artifact' of initial values but the result of them. This is similar to calibrating parameter values for which we just do not know the value. We agree that the model would be better tested if soil samples were known to be in steady-state conditions (an unlikely state after disturbance). We note that the same was applied during validation. (Note that in
20   semi-mechanistic models the outcomes are also strongly determined by the model structure).

*p. 11 l. 16: The results and subsequent discussion on p. 14 l. 14 of temperature effects on apparent Ea and Q10 is interesting. The authors should make clear that these effects were modelled from constant, temperature-independent values for Ea of different processes, so that these effects, which are widely*
25   *observed experimentally, are an emergent property of the model arising from offsetting effects on these processes (e.g. V vs. Km).*
We added lines (P 15  L 1-3) to make this idea clearer.

*p. 13 l. 18: 'where' to 'were'*
30   Done

*p. 16 l.2: Greater clarification of how diffusion in the model caused the very different water sensitivities of respiration with temperature would be helpful. Apparently diffusion became non-limiting to respiration at lower water contents with higher temperature, but why? Diffusivity does increase with temperature,*
35   *but this does not appear to have been modelled.*
This is a good question. As stated in the discussion (P 16 L 4-9), we are yet unsure and can only guess the underlying reasons. Clear causes are not always evident in complex models and a better explanation will require further model and hypothesis testing. Note, in any case, that the model is free to download and can be explored by anyone interested.
40
*p. 16 l. 12: Wouldn't CO2 concentrations of 56000 ppm suppress CO2 effluxes from the soil? Typically headspace air is replaced periodically during incubations*

Yes, the air was replaced each time measurements were taken. That value was the maximum for one sample with high rates, and less in all other cases. The text explains why these values were or were not significant for the results.

We rewrote the sentence to better reflect this.

[revised manuscript text omitted]